# Incipient diploidization of the medicinal plant *Perilla* within 10,000 years

Yujun Zhang [1,5✉], Qi Shen[1,2,4,5], Liang Leng [1,5], Dong Zhang[1], Sha Chen[1], Yuhua Shi[1], Zemin Ning[3] & Shilin Chen [1✉]

Perilla is a young allotetraploid Lamiaceae species widely used in East Asia as herb and oil plant. Here, we report the high-quality, chromosome-scale genomes of the tetraploid (*Perilla frutescens*) and the AA diploid progenitor (*Perilla citriodora*). Comparative analyses suggest post Neolithic allotetraploidization within 10,000 years, and nucleotide mutation in tetraploid is 10% more than in diploid, both of which are dominated by G:C → A:T transitions. Incipient diploidization is characterized by balanced swaps of homeologous segments, and subsequent homeologous exchanges are enriched towards telomeres, with excess of replacements of AA genes by fractionated BB homeologs. Population analyses suggest that the *crispa* lines are close to the nascent tetraploid, and involvement of acyl-CoA: lysophosphatidylcholine acyl-transferase gene for high α-linolenic acid content of seed oil is revealed by GWAS. These resources and findings provide insights into incipient diploidization and basis for breeding improvement of this medicinal plant.

[1] Institute of Chinese Materia Medica, China Academy of Chinese Medical Sciences, Beijing, China. [2] Rapeseed Research Institute, Guizhou Academy of Agricultural Sciences, Guiyang, China. [3] Wellcome Sanger Institute, Hinxton, UK. [4] Present address: Institute of Medical Plant Physiology and Ecology, School of Pharmaceutical Sciences, Guangzhou University of Chinese Medicine, Guangzhou, China. [5] These authors contributed equally: Yujun Zhang, Qi Shen, Liang Leng. ✉email: yjzhang@icmm.ac.cn; slchen@icmm.ac.cn

Polyploidization, or whole-genome duplication, is prevalent in angiosperms, and the resultant genetic redundancy is a key driver for species diversification, phenotypic innovation, environmental adaptation, and long-term evolution[1,2]. Recent evidence indicated that polyploidization has occurred much more frequently than estimation, involving most land plant lineages[3]. Polyploidization represented a genomic shock which resulted in gene expression deregulation, epigenetic instability, and meiotic difficulty[4,5]. A series of molecular events follows to meet these challenges for success polyploidization[2,4,5], and analysis of young polyploid species will provide insights into details of early diploidization at chromosomal, segmental, or nucleotide levels. For example, analysis of oilseed rape (*Brassica napus*, AACC genome) that formed about 7500 years ago revealed occurrence of homeologous exchanges, gene loss, and expression divergence between syntenic subgenomes[6]. In the past years, many polyploid genomes had been sequenced, such as peanut[7], strawberry[8], and sugarcane[9]. Unfortunately, most of these published polyploids were ancient, formed at least several million years ago, and our understanding of incipient diploidization is still limited.

Perilla is a recent allotetraploid species of the mint family Lamiaceae originated from China[10,11]. The plant is occasionally used as an ornamental bedding plant for its brightly colored red foliage. Perilla with frilly ruffled leaves, known as *shiso* in Japan, is widely used for culinary purposes. Popularity of Asian cuisine in recent decades has resulted in rising demand for perilla. Perilla had also been prescribed in Asian countries as a traditional herbal medicine. In addition, perilla is one of the plant species with the most abundant α-linolenic acid[12] (ALA). ALA is essential fatty acid for human that can only be acquired through diet[13], suggesting desirable health benefits of this plant. Classification of perilla has been done using morphological, agronomical, or chemical characters, often resulting in confused nomenclature,

since distinctions between varieties are ambiguous[14]. Karyotypically, the *Perilla* genus is composed of one tetraploid species *P. frutescens* ($2n = 4x = 40$) and one diploid species ($2n = 2x = 20$). It had been suggested that *P. citriodora* is a diploid donor for *P. frutescens*, while information on the second diploid ancestor is missing[10,11,14].

To better understand recent evolution of perilla since polyploidization, here we generate high-quality, chromosome-scale genome assemblies of *P. frutescens* and the diploid *P. citriodora*. Resequencing of 191 tetraploid accessions across China and abroad, as well as seven diploid lines, are used to extrapolate population structure and evolution at nucleotide, segmental, and chromosomal levels. Patterns and rates of nucleotide mutation since polyploidization are then measured. Finally, candidate genes for perilla leaf color variation and seed oil ALA content are identified by GWAS using high-resolution polymorphism data.

## Results

**Assembly of the perilla genomes**. An elite perilla cultivar PF40 with green leaves and high seed oil content (~56%) was selected for tetraploid genome assembly. Briefly, the *P. frutescens* (hereafter referred to as PF) genome size was estimated 1.24 Gb using K-mer frequency analysis (Supplementary Fig. 1), which agreed with the result from flow cytometry (1.12 Gb, Supplementary Fig. 2). A total of 54.5× coverage of single-molecule sequences of the PacBio Sequel platform was used for de novo assembly, and 130.0× Illumina data was generated for sequencing error corrections and gap filling. The final assembly was 1.241 Gb, with contig N50 of 3.21 Mb (Supplementary Table 1). We constructed high-throughput chromosome conformation capture (Hi-C) library to anchor scaffolds to chromosomes. Totally 54.7 Gb uniquely mapped valid Hi-C reads were used for scaffolding by LACHESIS software[15]. As a result, 1.203 Gb (97.5%) of the assembly were placed on 20 chromosomes (Fig. 1b,

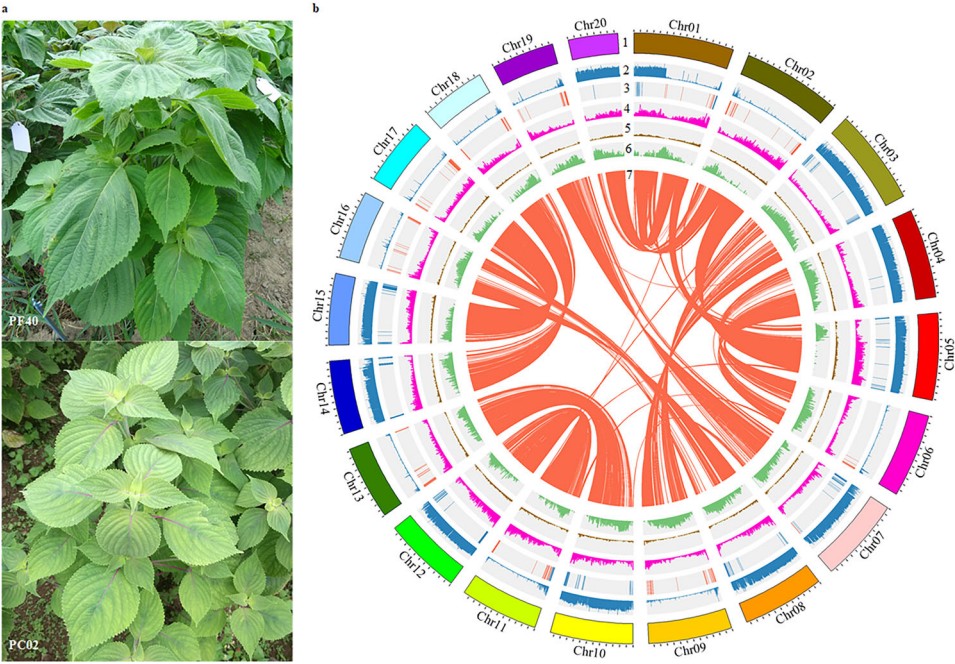

**Fig. 1 Genome of the allotetraploid *P. frutescens*. a** Images of mature plants of the allotetraploid PF40 and the diploid PC02 used for de novo assemblies. **b** Mapped features of the allotetraploid genome including (1) chromosomes arbitrarily numbered in descending order of their assembled lengths, (2) mapping depth distribution by PC02 in 10-kb windows, (3) distribution of 527 pairs of HE genes on PFA (as blue lines) and PFB (red lines) subgenomes, (4) density of predicted genes in 500-kb windows (with values 0–67), (5) density of predicted pseudogenes in 500-kb windows (0–67), (6) percentage of repeats in 500-kb windows (0.5–1.0), and (7) PFA-PFB synteny linked by red lines ($n = 15{,}170$). Ticks on the outer circumference represent 5-Mb units of chromosome length.

Supplementary Table 2, and Supplementary Fig. 3), with super-scaffold N50 of 62.64 Mb.

For diploid *P. citriodora* (hereafter referred to as PC), seven wild lines were first evaluated by resequencing and mapping onto the PF assembly (Supplementary Table 3). The obvious mapping dichotomy, where only half of the PF genomic regions were covered by these diploids (Supplementary Fig. 5), confirmed that PF is an allotetraploid, and all of the seven PC samples belong to the same diploid progenitor. We selected the least diverged sample PC02 for de novo assembly following the same PacBio and Hi-C procedures. The assembled PC02 genome is 676.9 Mb spanning 10 chromosomes, with super-scaffold N50 of 64.47 Mb (Supplementary Tables 1 and 4, Supplementary Fig. 3). The most diverged diploid PC99, being 10% smaller than PC02 in genome size, was assembled by Illumina approach for comparative analysis (Supplementary Table 5).

Heterozygosity of PF40 and PC02 are 0.16 and 0.10 SNPs per kb, respectively, about one-sixth of the out-crossing mint species *Mentha longifolia*[16], corroborating the selfing nature of the *Perilla* genus. On average, 96.18–99.05% of the Illumina paired-end reads (Supplementary Table 6) and 96.28–97.72% of the assembled transcripts (Supplementary Table 7) from published RNA-seq data[12,17] can be uniquely mapped to the genomes, while 92.08–92.71% of the 1440 genes in BUSCO evaluation dataset were completely covered by these genomes (Supplementary Table 8), demonstrating completeness of our assemblies.

We partitioned the PF genome into two nonoverlapping subgenomes. Segments with unique mapping coverage by PC02 were defined as AA diploid origin, and the remaining fragments were arbitrarily assigned to BB subgenome despite the absence of extant BB diploid species. Totally 634.6 Mb AA-derived sequences (hereafter referred to as PFA) were identified, similar to the size of PC02 genome. Taking into account of the ~99% unique mapping rate of PC02 sequencing reads to PF genome, it suggested that most of the sequences from AA diploid donor species had been kept in the tetraploid genome. It is noteworthy that chr1, chr7, chr11, and chr20 were all chimeric with both AA- and BB-derived segments (Supplementary Table 9), suggesting inter-subgenome reshuffling during diploidization.

**Repeat and gene annotation**. We identified transposable elements (TEs) that contributed 64.1% and 56.7% to PF and PC02 genomes, respectively (Supplementary Data 1). Long terminal repeat (LTR) retrotransposon is the most abundant type of repeats in perilla, occupying 22% of the genomes, and contents of *copia* and *gypsy* LTRs are about the same within PFA, PFB, and PC02 sequences (Supplementary Fig. 6). Further analysis indicated that most LTRs were shared between PC02 and PFA (3786 LTRs, Supplementary Fig. 7), while the number of LTRs emerged in tetraploid (659) was slightly more than that in diploid (507).

We annotated protein-coding genes of the *Perilla* genus by ab initio prediction, homology-based prediction, EST alignment, and RNA-seq assembly. Availability of four syntenic sequences (PFA, PFB, PC02, and PC99) enabled comparative gene model curation, which in turn facilitated identification of pseudogenes formed after polyploidization. For this purpose, we first constructed syntenic relationship among the four sequences by aligning draft gene models with Mercator[18], the resultant 4,030 collinear blocks were then aligned by MAFFT[19], spanning 473 and 468 Mb sequences of PFA and PFB, respectively. In most cases, four syntenic gene models were observed within each orthologous interval, and scores of RNA-seq support and homologous protein hit from GenBank were used for evaluation. Finally, the best gene model was selected as standard gene prediction and projected onto the other three orthologous sequences by GMAP[20] for comparative curation. Totally 23,549, 19,978, 25,662, and 23,819 genes were annotated for the four genomes, respectively. Specifically, there were 22,865 unique ancestral gene models across the syntenic chromosomal regions of PFA, PFB, and PC02 (Supplementary Fig. 8). PFA had 666 pseudogenes because of premature stop codons or frameshifts and 704 gene deletions within the syntenic intervals, while PFB had 1510 pseudogenes and 4473 gene deletions. This asymmetrical gene loss between homeologous chromosomes ($P < 2.2 \times 10^{-16}$, Chi-squared test), frequently referred to as 'genome fractionation'[4], implied that AA was the dominant subgenome. It is noteworthy that pseudogene identification by comparative curation resulted in slightly more missing gene models in BUSCO evaluation. Detailed sequence inspections suggested that about one-third of these retrieved pseudogenes were caused by heterozygous coding SNPs or Indels in the three genomes, with the remaining pseudogenes by fixed coding variations (mainly Indels, Supplementary Table 8).

We analyzed gene families with protein sequences from ten published plant genomes and the four perilla sequences (Supplementary Data 2, Supplementary Table 12). Totally 24,331 families were built, ranging in size from 2 to 1753 genes. Phylogenetic tree with 606 single-copy orthologous genes suggested that the *Perilla* genus was closely related to *Salvia*, both of which belong to Lamiaceae family. For perilla speciation, the AA diploid was first diverged from BB about 2.3 million years ago (Mya), then PC02 separated with PC99 about 0.8 Mya, and a later hybridization of PC02 with a yet unknown BB donor gave rise to the allotetraploid 0.2 Mya (Supplementary Fig. 9).

We further calculated divergence between four perilla sequences with syntenic orthologous genes. Distributions of pairwise synonymous substitution rates ($Ks$) of the three sets of AA-BB gene pairs all peaked around 0.034 (Fig. 2a). Assuming an average plant mutation rate of $7.1 \times 10^{-9}$ substitutions per synonymous site per year[21], it implied that the two diploid progenitors diverged about 2.4 Mya, close to the estimation based on single-copy genes. Surprisingly, coding sequences of 8939 orthologous genes between PFA and PC02 had no synonymous substitutions ($Ks = 0$, 49.1%), and 5617 gene pairs among them even had identical coding sequences (30.9%), resulting in exponential decay of $Ks$ distribution plot with no peak. Indeed, 260 out of the 606 single-copy orthologous genes had no synonymous substitutions either, implying that molecular dating by concatenating coding sequences of single-copy genes over-estimated polyploidization time in this extreme scenario[22]. This is corroborated by 71 shared LTR-RTs between PFA and PC02 that had identical pairwise sequences at long terminal ends, while variations between PFA and PC02 were as low as 1.9 SNPs per kb in exonic regions on average (Supplementary Table 13). Indeed, the estimated age of perilla allotetraploidization was only one-third of that for *Brassica napus* based on single-copy genes (Supplementary Fig. 9). Compared with the ~7500-year-old allopolyploid *Brassica napus* where 18.6% genes were identical between tetraploid and diploid progenitor[6], the allotetraploid *P. frutescens* should have formed post Neolithic within the recent 10,000 years, providing an ideal plant species to elucidate incipient polyploid evolution at sequence level.

**Recent polyploid evolution**. Allopolyploid speciation represents a genomic shock which requires rapid evolutionary reconciliation of two diverged genomes and gene regulatory networks[5]. To reveal molecular details of incipient diploidization of perilla, we first analyzed genome synteny between the two species. As expected, each PC segment has two syntenic PF counterparts (Fig. 2b). Large-scale variations of BB-derived chromosomes, especially chr2, chr6, chr16, and chr19, were observed when

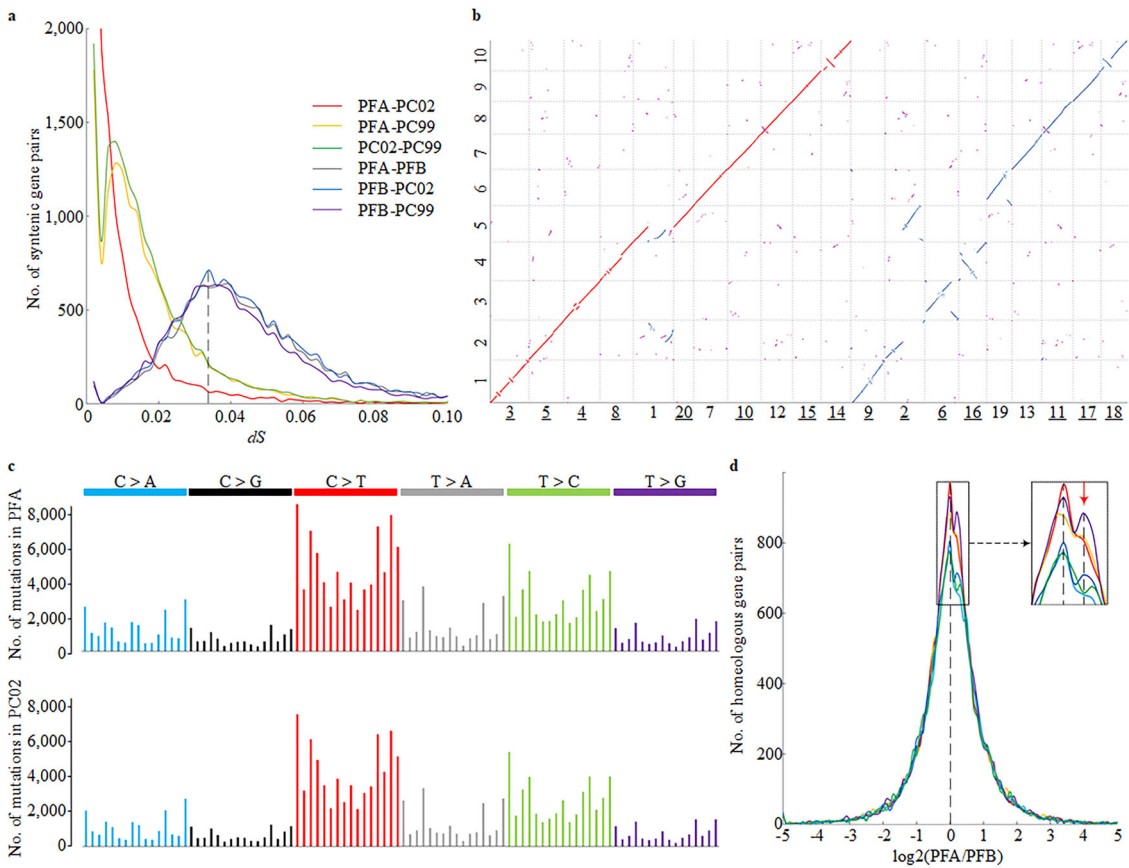

**Fig. 2 Evolution of the allotetraploid *Perilla*. a** Distribution of synonymous nucleotide substitutions (*dS*) between the four perilla sequences. The *dS* = 0 signal between PFA-PC02 (*n* = 8939) was not displayed. **b** Chromosomal synteny between PF and PC genomes. Each dot represented syntenic gene relationship between PFA-PC02 (19,412 gene pairs, in red) or PFB-PC02 (15,422 gene pairs, in blue). Scattered segmental duplications not related to polyploidization were shown by magenta dots. PF chromosomes underlined were reversed for visual consistence. **c** Patterns and statistics of nucleotide mutational signatures of PFA and PC02 since polyploidization. The signatures are displayed according to the 96-substitution classification defined by substitution class and sequence context immediately 5′ and 3′ to the mutated base, and displayed alphabetically from ANA to TNT. **d** Subgenome expression dominance as calculated by log2 transformed TPM (Transcripts Per Million) ratio of PFA to PFB syntenic genes (*n* = 15,484). Solid lines represented RNA-seq data of PF40 from flower and leaf with three replicates each. For any paired TPM values of <1, a pseudo-count of 1 was added to both PFA and PFB values before log2 ratio calculation. Enlarged inset showed expression bias toward PFA with a minor peak around 0.2 (red downward arrowhead). Source data underlying Fig. 2c are provided as a Source data file.

compared with PC/PFA. With the shortage of the BB donor and outgroup information, directions and dating of these structural variations cannot be determined, yet the chimeric structure of chr1, involving at least three breakages and two fusions between AA and BB segments (Supplementary Fig. 10), must postdate allotetraploid formation. While LACHESIS is known to introduce large-scale inversion errors occasionally, detailed sequence alignments suggested 17 out of 19 chromosomal inversions were authentic which were validated by Illumina draft assembly data (Supplementary Data 3). Three bona fide inversions of PC segments were observed, with the largest one of 14.8 Mb supported by micro-synteny with Arabidopsis (Supplementary Fig. 11, Supplementary Table 14), suggesting that the diploid donor species was also in dynamic karyotype evolution since polyploidization. Further analysis of PC02 genome revealed 66 scattered duplication blocks involving 1812 pairs of genes (Supplementary Fig. 12), representing relics of ancient whole-genome duplication history of the diploid. The calculated *Ks* distribution peak around 0.9 had been observed in most Lamiaceae species[23,24], suggesting the occurrence of ancient polyploidization at the basal lineage of Lamiaceae about 68 Mya[23].

As Haldane put in 1932 that "(evolution) will favor polyploids, and particularly allopolyploids, which possess several pairs of sets

of genes, so that one gene may be altered without disadvantage"[25], polyploid species will have elevated mutation rate when compared with diploid progenitors. Taking advantage of MAFFT alignment of four homologous sequences, we reconstructed ancestral nucleotide for each conserved position to calculate mutation rate and direction in both tetraploid and diploid since polyploidization. To reduce ambiguity from erroneous multiple alignments, we focused on syntenic segments longer than 500 bp with overall sequence identity higher than 80%, and only those 1:3 genotype patterns were counted as one mutation and three ancestral nucleotides (Supplementary Fig. 13). A total of 42,022 syntenic blocks spanning 123.6 Mb sequences were analyzed. Missing of BB diploid sequences compromised our ability of discerning ancestral genotype between AA and BB, but had no effect on mutations between AAs of diploid and tetraploid. It turned out that frequencies for all of the six types of nucleotide mutations were about the same between PFA and PC02, with transition/transversion ratio of 1.8 (Fig. 2c and Supplementary Table 15). G:C → A:T transitions were by far the most frequent type of mutations, accounting for 40% in both PFA and PC02. This is comparable to the abundant G:C → A:T substitutions observed in single-seed descending Arabidopsis lines after 30 generations[21] and de novo mutations in human

somatic cells[26], supporting the notion that the mutated thymines are derived from spontaneous deamination of methylated cytosines. Enhanced cytosine methylation level in newly formed polyploid plants[27] may reinforce this mutational bias. Collectively, we identified 206,069 and 187,181 de novo mutations on the syntenic AA sequences of tetraploid and diploid, respectively (Supplementary Table 15). The 10% excess of nucleotide mutations ($P = 0.01$, one-side paired $t$-test), as a quantitative measurement of polyploid masking effects, confirmed a relaxed selective pressure on polyploid species during evolution.

Transcriptional adaptation to polyploidization usually ranges from complete suppression of transcription from one genome to down- or upregulation of gene expression[5]. We evaluated homeolog expression bias of 15,484 syntenic PFA-PFB gene pairs. It seemed that the PFA/PFB expression ratio followed symmetrical distribution to some extent, with a minor peak around 0.2 (equivalent to TPM ratio of 1.149), suggesting slight upregulation of PFA genes (Fig. 2d). This agreed with the subgenome-dominance hypothesis[28], where the dominant AA subgenome had greater gene content and contributed more to global transcriptome than the fractionated BB subgenome.

While meiotic recombination in plants predominantly occurs between allelic sequences along homologous chromosomes, merging of two divergent genomes in one nucleus enables transfer of genetic information between paralogous or homeologous sequences because of the reduced pairing fidelity during meiosis of nascent polyploid[29]. Based on mapping depth analysis of population resequencing data, we identified replacement events of chromosomal regions with duplicated copies from corresponding homeologous sequences, or homeologous exchanges (HEs)[6]. Totally 29 HEs were identified (Fig. 3a, Supplementary Data 4), ranging from 17 to 595 kb in size. It is noteworthy that the 15 PFA replacements by PFBs were observed totally 525 times across the 191 resequencing samples, while the 14 PFB replacements by PFAs occurred only 103 times, implying that early HEs were dominated by PFB duplications which were subsequently shared by more varieties ($P = 3.5 \times 10^{-5}$, Chi-squared test). We further detected 527 HEs at genic level (or gene conversions). Intriguingly, 314 of them (59.6%) located proximally to chromosomal ends within 2 Mb which harbored 11.7% of total genes (Fig. 1b). Forty-six bi-directional genic HEs with duplications of PFB in some accessions and PFA in others were also identified (Supplementary Data 5). Enrichment of genic HEs toward telomeres ($P < 2.2 \times 10^{-16}$, exact binomial test) might result from initiation of homeologous chromosome contact from telomeres through the bouquet configuration during meiosis[30]. Indeed, higher conversion rate for genes proximal to telomeres had been observed in *Brassica rapa*[31], *Brassica napus*[32,33], and nascent allopolyploid wheat[34,35], suggesting a common response to the challenges of merging two divergent genomes in one nucleus.

By mapping PC reads onto PF genome, we identified 18 balanced reciprocal translocations between syntenic PFA and PFB homeologs (Fig. 3b, Supplementary Table 16). These reciprocal DNA exchanges had been inherited by all tetraploid accessions, and can only be observed by mapping diploid reads, suggesting earlier occurrence than homeologous exchanges.

In addition, nonreciprocal transfer of genetic information between non-homeologous sequences was observed in three lines, which bore classic signature of gene conversion in meiosis[36] (copy ratio of 3:1, Supplementary Data 6). Two of these events resulted in gene dosage imbalance of ~35 Mb segments (Fig. 3c), and Illumina reads spanning the exchange junctions confirmed the authenticity of these aberrations (Supplementary Fig. 14), suggesting de novo aberrant exchanges during meiosis of nascent polyploids.

**Genomic variation and population structure**. We selected 191 tetraploid accessions representing all perilla taxonomical groups from China and abroad for whole-genome resequencing on Illumina HiSeq platform. A total of 4135 Gb clean data were generated, equivalent to about 16× sequencing coverage for each line (Supplementary Data 7). We identified 11,737,374 single-nucleotide polymorphisms (SNPs) and 2,070,111 indels. Totally 3442 nonsense SNPs and 12,214 frame-shift indels affecting 8592 protein-coding genes were annotated, highlighting genetic basis of phenotypic diversity of perilla population. There were 27 accessions with relatively higher heterozygosity (Supplementary Fig. 15b), suggesting possible history of interbreeding.

To clarify taxonomy of these accessions, we performed phylogenetic-tree construction (Fig. 4a), principal component analysis (PCA, Fig. 4b), and population-structure analysis (Fig. 4c) with 4.8 M filtered SNPs. It turned out that these accessions can be classified into three clades, var. from south China, var. from north China, and var. *crispa*, including 120, 26, and 45 accessions, respectively. *Crispa* perilla is characterized by wrinkled and curled leaves with rounded serrated edges, and had been reported occasionally from Japan, Korea, China, Myanmar, and Vietnam, suggesting a wide distribution from East to Southeast Asia. Taxonomical abundance of the south China group, on the other hand, indicated dense sampling of accessions from Guizhou Province, a major perilla habitat in southwest China. Genetic diversity ($\theta\pi$) value was $1.00–1.45 \times 10^{-3}$ across the three clades, and population-differentiation statistic ($F_{ST}$) among the three groups was 0.175–0.223, with var. *crispa* highly diverged with the other two clades (Fig. 4d). The average pairwise correlation coefficient ($r^2$) of linkage disequilibrium (LD) decayed to half maximum at 175 kb for south China group and var. *crispa*, while the north China group's LD block extended up to 900 kb (Supplementary Fig. 16).

We used PFA-specific SNPs against PC02 from MAFFT alignment to further clarify perilla phylogeny. Totally 108,850 out of 206,069 SNPs (52.8%) were polymorphic within the resequencing population, suggesting that these SNPs had not been fixed in tetraploid yet. Together with PC02 (having ancestral genotype at every position) and PF40 (having alternative allele at every position), a neighbor-joining tree was constructed by focusing on the AA subgenome only. Intriguingly, PC02 was placed in the center of the *crispa* clade (Fig. 4e), indicating that *crispa* represented the nascent tetraploid perilla lineage. Since the reference accession PF40 belonged to south China clade, the seemingly abundance of variations of the *crispa* lines (Supplementary Fig. 15c), on the contrary, implied their least divergence with the PC diploid progenitor. We retrieved Illumina reads that mapped on PFA sequences, and re-mapped them onto PC02 to calculate divergence of the three tetraploid clades against the diploid. It turned out that the *crispa* lines had more conserved ancestral nucleotides and less fixed mutations than north China and south China lines (Supplementary Fig. 15d), corroborating the basal position of *crispa* perilla in tetraploid phylogeny[37].

**Genome-wide association studies of key agronomic traits**. We used the high-density polymorphism map of perilla to elucidate genetic basis of key agronomic traits by genome-wide association study (GWAS). Perilla has two chemo-varietal forms: the anthocyanin-producing red forma which is widely used as food coloring, and the green forma as receptacle to hold sushi wasabi in Japanese cuisine. By mapping leaf color phenotype onto neighbor-joining tree, it turned out that most red lines were observed within the *crispa* clade (26/32, Fig. 4e), while all south China clade accessions have green leaves including the reference

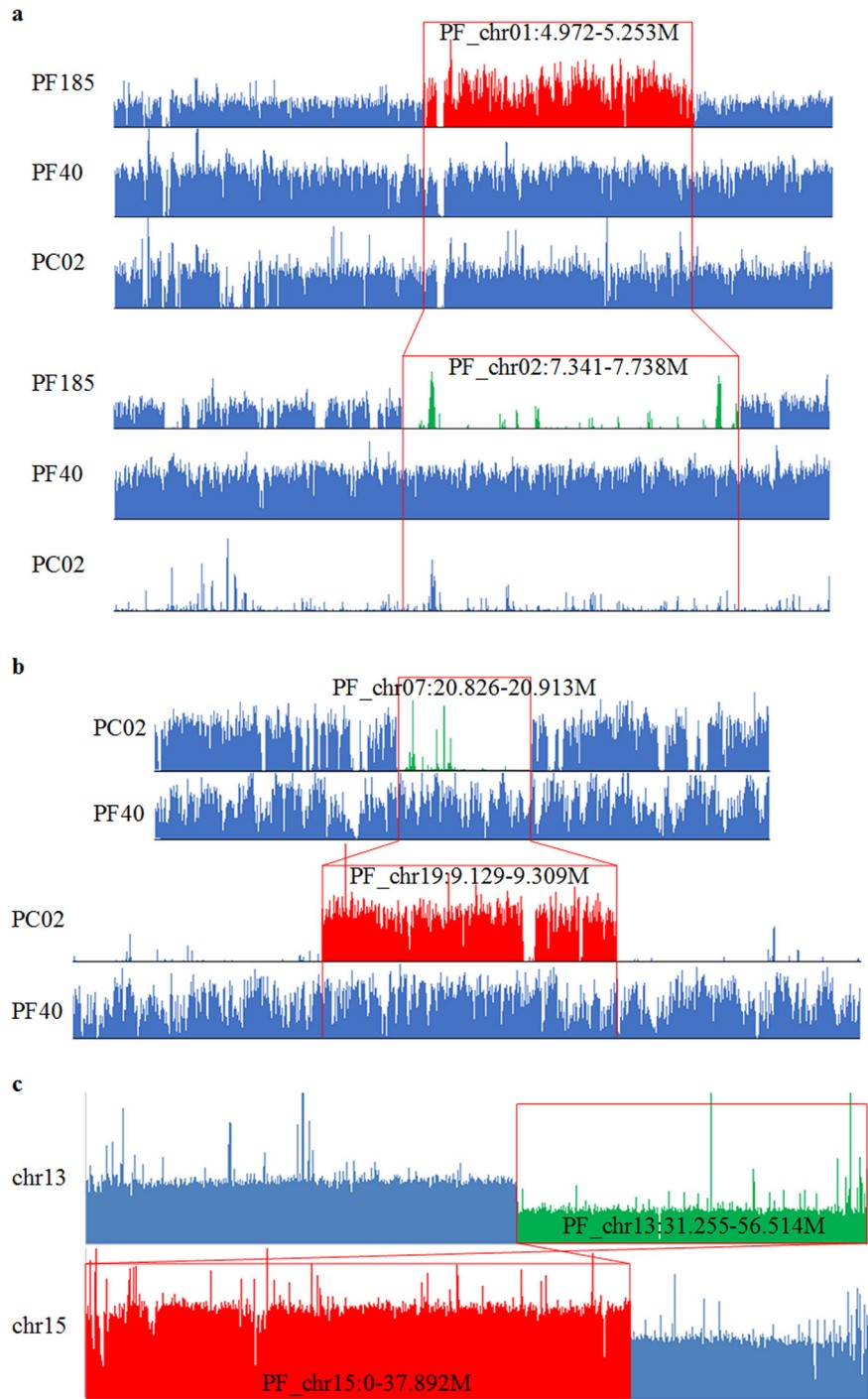

**Fig. 3 Patterns of subgenome exchanges in the allotetraploid. a** Sequencing depth distribution between syntenic subgenomes indicated homeologous exchange in PF185. Coverage by PC02 indicated that the chr1 block was of AA-origin. In PF185, a 397-kb segment of chr2 (deletion, shown in green) was replaced by 281-kb homeologous sequences of chr1, resulting in a seemingly duplication of the chr1 segment (in red). Red boxes marked the corresponding HE intervals. **b** Reciprocal HE swapped 180 kb of chr7 with its syntenic chr19 interval. This balanced exchange was inherited by all tetraploid accessions. **c** Replacement of chr13 segments by non-homeologous chr15 sequences resulted in gene dosage imbalance (1:3) in PF175.

line PF40. GWAS analysis within *crispa* (26 red *vs*. 19 green) gave a strong signal toward a predicted Myb transcription factor gene on chr8 ($P = 1.05 \times 10^{-7}$ after Bonferroni correction, Fig. 5a), and micro collinearity with Arabidopsis unequivocally suggested a Myb113 homolog (Supplementary Fig. 17). As expected, the reference Myb113 is a pseudogene, and three coding variants were identified in green-leaf accessions, including 6-bp in-frame deletion in the 2nd exon, 1-bp deletion, and one nonsense SNP in

the third exon (Fig. 5b), while all red lines had kept intact Myb113 proteins. MYB113, together with its homologs MYB114, MYB75/PAP1, and MYB90/PAP2, constitute the four R2R3-type MYB transcription factors that regulate anthocyanin biosynthesis in plant vegetative tissues by forming MYB-bHLH-WD40 complex[38]. PAP1/2 had implicated in leaf color variation in cotton[39] and Arabidopsis[40], and overexpression of Myb113/ Myb114 resulted in increased anthocyanin accumulation in

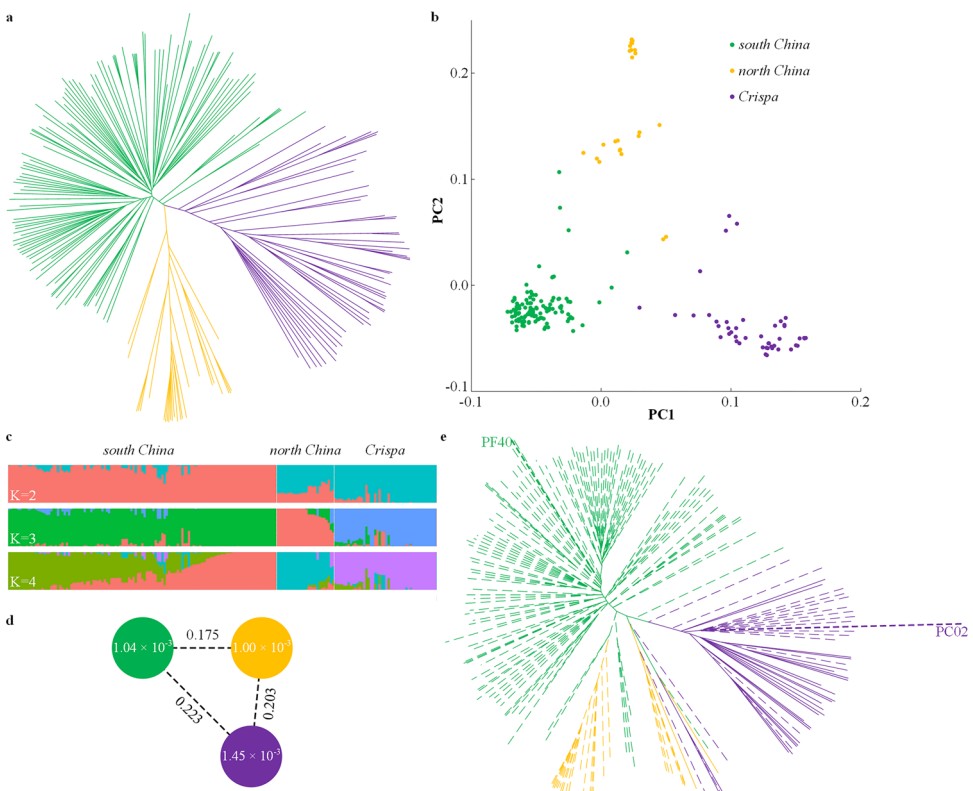

**Fig. 4 Population analysis of perilla germplasms. a** Neighbor-joining tree of 191 allotetraploid perilla accessions. Green, yellow, and purple branches indicated three perilla clades of south China ($n = 120$), north China (26), and *crispa* (45), respectively. Totally 4,789,738 filtered high-quality SNPs were used for tree construction. **b** PCA analysis of these samples with the same SNP dataset. **c** Population structure of these samples based on different numbers of clusters ($K = 2$–4). **d** Nucleotide diversity ($\pi$) and population divergence ($F_{ST}$) across the three clades. The value in each circle indicates calculated nucleotide diversity for that clade, and the value along each line represents population divergence between two neighboring clades. **e** Neighbor-joining tree constructed with 108,850 PFA-specific SNPs, and overlaid with leaf color information: dashed lines indicated green leaves, and solid lines represented red leaf phenotype. Color codes were the same in (**a**), (**b**), (**d**), and (**e**).

Arabidopsis[41,42], supporting the pivotal role of Myb113 in perilla leaf coloration.

It is noteworthy that three *crispa* accessions and seven diploids, all with intact Myb113 copies, had green leaves, suggesting additional factors for color expressivity. Detailed sequence inspection revealed no loss-of-function mutations on anthocyanin biosynthesis genes[17] in these samples, while a 9967-bp fragment deletion upstream of Myb113 was identified in all red lines, which is the 3′ segment of a *gypsy* type LTR element (Fig. 5c). We crossed a red line PF899 with a green line PF084, both of which had intact Myb113 but PF084 had no upstream deletion. It turned out that leaves of the F1 plants are pale red, and red leaves of the F2 lines co-segregated with the deletion ($n = 110$). Taking into account of perilla phylogeny, this result suggested that red leaf phenotype was disabled in AA diploid by retrotransposon insertion before polyploidization, which was then restored by partial removal of LTR toward 3′ end in *crispa* clade of tetraploid. Green *crispa* lines with intact LTR accumulated loss-of-function mutations on Myb113 over time, leading to emergence of the north and south China clades (Fig. 5d). Intriguingly, a similar pseudogenization and resurrection scenario of R2R3-MYB transcription factor for floral color evolution had been reported in *Petunia secreta* recently[43]. Insertion and removal of upstream LTR presumably regulated Myb113 expression (Supplementary Fig. 18a), which needs further investigations.

In developing plant seeds, fatty acids are exclusively synthesized in plastids, and the nascent fatty acids, mostly as oleic acid (C18:1), palmitic acid (C16:0), and stearic acid (C18:0), are exported to cytoplasm to enter into the acyl-CoA pool. C18:1 in cytoplasm is then esterified to the phosphatidylcholine at *sn*-2 position by acyl-CoA:lysophosphatidylcholine acyltransferase LPCAT[44]. C18:1 was subsequently desaturated into LA (C18:2) and ALA (C18:3) by fatty acid desaturases FAD2 and FAD3 on endoplasmic reticulum, respectively. Phosphatidylcholine is the only site for ALA synthesis in plant seeds, and the polyunsaturated fatty acids on the *sn*-2 position of phosphatidylcholine were transacylated onto *sn*-3 position of diacylglycerol by phospholipid:diacylglycerol acyltransferase, resulting in production of the energy storage lipid triacylglycerols (TAGs)[45]. We annotated these fatty acid genes (Supplementary Data 8) and analyzed their expression during seed development. It turned out that expressions of microsomal FADs were upregulated from 2 days post anthesis (DPA2) and peaked from DPA14, with a similar pattern of the oleosin family genes from DPA10 (Fig. 6a), suggesting that the high LA and ALA content of perilla were mainly correlated with elevated transcriptional levels of FAD2/FAD3 and oleosin genes of lipid body.

We performed GWAS analysis for seed ALA content in our sample collections, which varied from 47.9 to 66.5% of TAGs in our germplasm collection. A strong signal on chr2 near an LPCAT homolog was identified ($P = 2.95 \times 10^{-5}$ after Bonferroni correction, Fig. 6b). Biochemical coupling of LPCAT and diacylglycerol acyltransferase had been revealed to contribute the enriched polyunsaturated fatty acids incorporation into TAG in flax[46], another plant species with high ALA content,

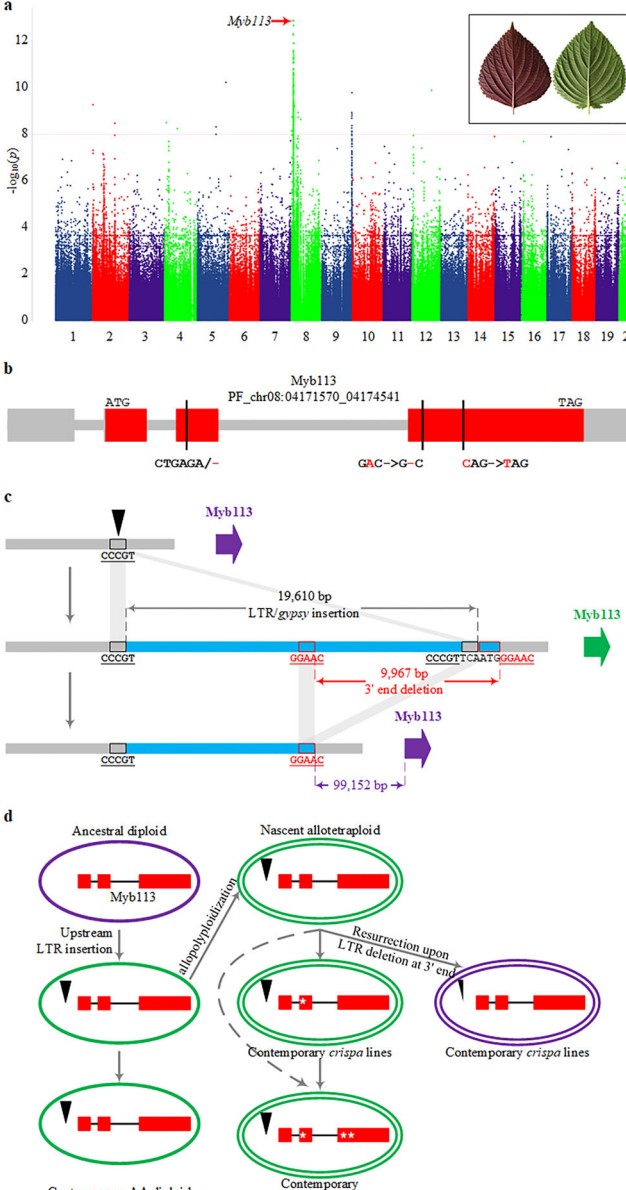

**Fig. 5 GWAS analysis of perilla leaf color variation. a** Manhattan plot. The strongest associated SNP on chr8 was marked by a red arrow. Dashed horizontal red line indicated significant *p*-value threshold of 1.04e−8 (calculated as 0.05/n). Leaf (abaxial) phenotypes were shown as inset. **b** Exon structure of the causal Myb113 gene. Coding exons are shown as red boxes, and UTRs in gray. Positions of three coding variants are marked by vertical black lines. **c** Schematic representation of LTR insertion in diploid and subsequent 3′ end partial deletion in allotetraploid. Black box on chromosome (gray line) indicated the 5-nt target site that was duplicated upon LTR integration, and red box on chromosome represented the 5-nt micro-homologous sequences that initiated the 9967-bp segment deletion. Drawn not to scale. **d** A proposed scenario for evolution of perilla leaf color. Note that the 6-bp in-frame deletion in the 2nd exon of Myb113 (white asterisk) first emerged within the *crispa* clade.

corroborating involvement of LPCAT in ALA accumulation in oilseeds. No causal variants were observed in LPCAT, while a 40 kb fragment deletion spanning GWAS peak interval tagged ALA content well (Fig. 6c), suggesting that the reduced ALA content of the deletion lines might result from transcriptional regulation of LPCAT (Supplementary Fig. 18b).

## Discussion

It had been widely accepted that nearly all extant angiosperm genomes contain vestiges of multiple rounds of polyploidy. Immediately after polyploidization, nascent polyploid must pass through a bottleneck of genomic disruption[5,47], including changes in cellular architecture, difficulties in meiosis, regulatory changes of gene expression, and alteration of epigenetic landscapes, before becoming adapted and fueling long-term diversification. Recent analysis of newly formed natural or resynthesized allopolyploids, such as Brassica[32,33], wheat[34,35], Tragopogon[48], cotton[49], and monkeyflower[50], had revealed extensive inter- and intragenomic rearrangements, homeologous exchanges, subgenome expression dominance, deletion/silencing of TEs, and meiotic irregularities, representing major genetic processes accompanying nascent allopolyploidy. As a young allotetraploid species of 10,000 years old, perilla provided a unique opportunity to understand incipient diploidization. Asymmetrical evolution between perilla's subgenomes was conspicuous, including more intrachromosomal rearrangements of PFB than PFA, higher gene retention and expression, and low pseudogenization of PFA than PFB, and excess of homeologous replacements of PFA genes by PFB homeologs.

Recombinations between homeologs will surely contribute to intraspecific diversity and adaptation[6,51,52]. However, recurrent HEs toward telomeres, as we identified here in perilla, can also increase the global genomic similarity between homeologous chromosomes, leading to even more illegitimate crossovers, unequal bivalents, and inviable gametes, thus being detrimental to successful establishment of polyploid. On the contrary, balanced swap of homeologous segments can maintain genomic divergence, prevent illegitimate pairing, thus facilitate nascent polyploid stabilization. It is noteworthy that frequencies of perilla HEs varied from 0.5 to 45.0% (Supplementary Data 4), while the 18 balanced exchanges were shared by all polyploid lines (Supplementary Table 16), suggesting that the early occurred balanced swap of homeologous segments is critical for incipient diploidization.

Since suppression of homeologous pairing during meiosis is essential for correct chromosome partitioning, it had been suggested recently that meiosis of nascent allopolyploid can be stabilized through abolishment of non-homologous crossovers[53] by reducing MSH4, a key meiotic recombination gene, to single copy. We annotated essential genes involved in homologous chromosome synapsis and crossover formation in plants. It turned out all of these genes had four copies (Supplementary Data 9), corroborating the genetic basis of perilla's proneness to homeologous exchange and aneuploidy. Indeed, the full spectrum of meiotic crossover products between subgenomes had been observed in perilla, from 4:0 (classical HEs, Fig. 3a), 2:2 (balanced swap, Fig. 3b), to 3:1 (nonreciprocal exchange, Fig. 3c). Restoring to single copy of MSH4 will presumably result in more stable diploidized perilla accessions.

Similar to the post-Neolithic evolution of allopolyploid *Brassica napus*[6], our analysis of the perilla genomes revealed more details of recent polyploidization. The high-quality genomes and dense polymorphism map of perilla, on the other hand, will facilitate identification of key genes for agronomical and chemical traits (Supplementary Data 10). Taking together, these resources and findings provide a foundation for further understanding of incipient diploidization, and for genetic improvement of perilla and other Lamiaceae species.

## Methods

**Sample collection.** The tetraploid sample PF40 ($2n = 4x = 40$) was first collected from Huaxi District of Guiyang City, Guizhou Province (26°21′N, 106°32′E), and maintained in greenhouse for more than five successive generations by

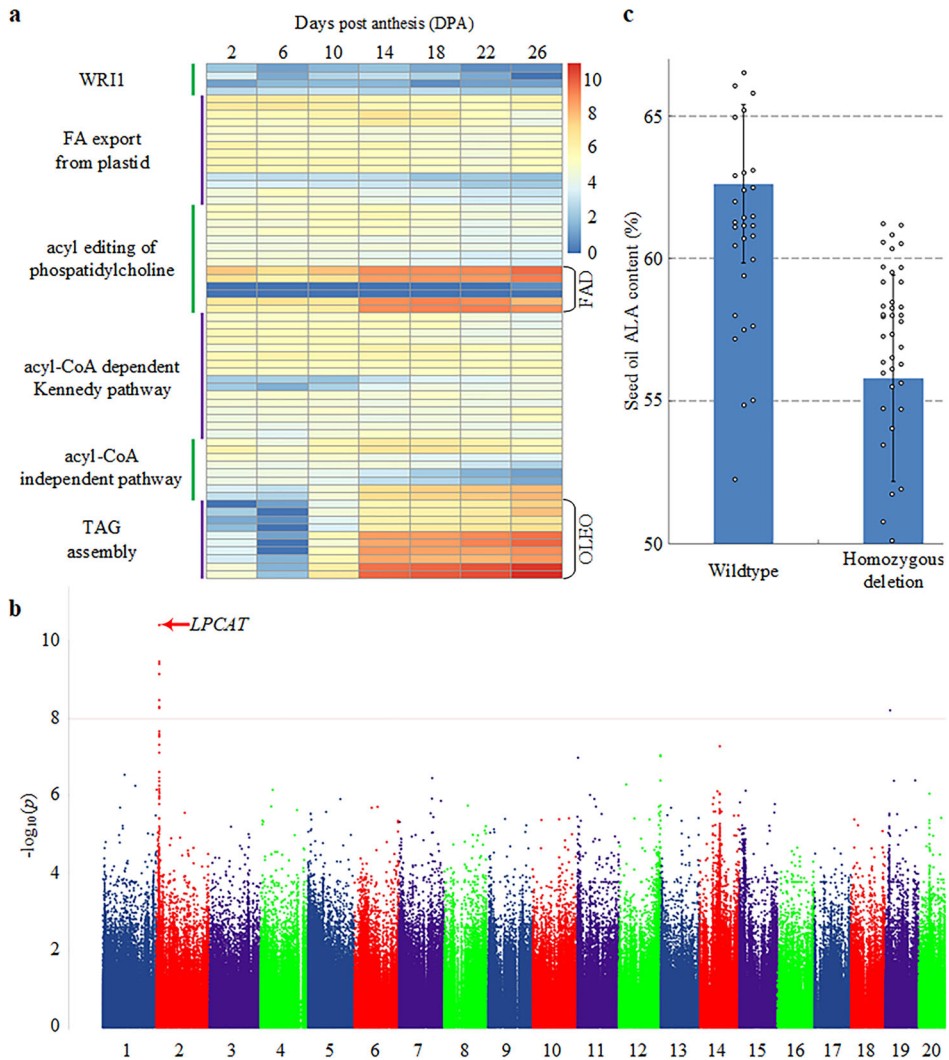

**Fig. 6 Analysis of perilla seed oil biosynthesis genes. a** Expression heatmap of TAG biosynthesis genes during perilla seed development. TPM values of TAG genes were extracted from seven transcriptomes corresponding to the seeds of 2, 6, 10, 14, 18, 22, and 26 days post anthesis (DPA) of the high oil content line PF40 (~56%), and displayed as log2(TPM + 1). Each row represented one predicted TAG-related genes in PFA-PFB alternating manner. The first column indicated functional categories of these genes. A detailed list of these 33-pairs of syntenic genes can be found in Supplementary Data 8. **b** Manhattan plot of GWAS analysis for ALA content of perilla seed oil. **c** Comparison of ALA content between two SV haplotypes in the GWAS population. Error bars, mean ± s.d. Source data underlying Fig. 6a are provided as a Source data file.

self-pollination. We selected this green-leaf accession for whole-genome de novo assembly because of its superior characters, including high grain yield and high seed oil content. The two diploid samples PC02 and PC99 ($2n = 2x = 20$) were both collected from Tianmu Mountain Nature Reserve of Zhejiang Province (30°21′N, 119°26′E), a known region with high perilla germplasm diversity[10]. A single plant from each of these three materials was used for genomic DNA extraction and sequencing. Fresh leaves were harvested and frozen immediately with liquid nitrogen, and high-molecular weight genomic DNA was extracted with the standard cetyltrimethyl ammonium bromide (CTAB) method[54]. DNA was then assessed by agarose gel electrophoresis and Agilent 2100 Bioanalyzer for quality and concentration, and finally purified with QIAquick Gel Extraction kit (Qiagen) for subsequent sequencing library construction.

**Library construction and sequencing**. For Illumina data generation, short-insert libraries (500 bp, 800 bp) were constructed with TruSeq DNA Library Prep Kit (Illumina), and mate-pair libraries (2, 5, and 10 kb) were constructed with Nextera Mate Pair Library Prep Kit (Illumina). Sequencing was run on HiSeq 2000 platform with PE125, PE150, or PE250 mode (Supplementary Table 6). Linked Reads libraries for PF40 and PC99 were further constructed with the Chromium platform[55] (10X Genomics) and sequenced. For PacBio sequencing, standard DNA Template Prep Kit 3.0 (Pacific Biosciences, USA) was used to prepare PacBio SMRTbell libraries of 20-kb insert size, followed by sequencing on PacBio Sequel platform using P6-C4 chemistry (Novogene, Beijing). Totally 67.6 and 38.9 Gb raw data were generated for PF40 and PC02, respectively. One Hi-C library was

prepared and sequenced on Illumina HiSeq 2000 to generate 54.7 and 21.8 Gb uniquely mapped valid Hi-C reads for PF40 and PC02, respectively.

**Genome assembly**. We initially chose the Illumina procedure to assemble PF40, PC02, and PC99 genomes with a combination of different Illumina assemblers (Supplementary Fig. 4a). Raw sequencing reads were processed to screen out low-quality data, and contig-only assemblies were generated by both Fermi[56] and Phusion2[57]. SOAPdenovo[58] was used independently for assembly, which was then improved using SSPACE[59]. We then used the Fermi/Phusion2 assemblies to replace contig sequences from SOAP assembly to improve accuracy of indels, while scaffold structure was kept intact. To further improve the draft assemblies, long linked-reads from 10X Genomics were used for scaffolding with Scaff10X pipeline (https://www.sanger.ac.uk/science/tools/scaff10x), resulting in the Illumina versions of PF40, PC02, and PC99 genome assemblies.

The fragmented nature of these Illumina assemblies, with contig N50s of <100 kb, limited our analytical resolution on incipient diploidization of perilla. For this reason, we re-assembled the PF40 and PC02 genomes by PacBio/Hi-C procedures using the same perilla lines. PacBio sequencing data were first assembled with Canu[60] v1.5, and only reads longer than 1 kb were used. The assembled genomes were corrected by Pilon[61] v1.20 using Illumina paired-end data for two rounds. Hi-C sequencing data were aligned to the consensus contigs by Bowtie2[62], then processed by Hi-C-Pro[63] v2.7.8, and finally agglomerative hierarchical clustering by LACHESIS was used to generate the chromosomal maps of PF40 and PC02. With the shortage of physical map information of the two

species, chromosomes were arbitrarily numbered in descending order of their assembled lengths.

To evaluate consistency of the two assembly versions, we first cut the Illumina data of PF40 into pseudo mate-pair sequences spanning 1, 5, 10, and 20 kb, respectively, with read length of 150 bp, and mapped onto the PacBio version by BLAST[64] (v2.2.28+, BLASTN). Mapping distance of the top1 hit (≥99% similarity and ≥95% query coverage) and configuration of the mate pair were used for evaluation (Supplementary Fig. 4b). Second, the two PF40 versions were pairwisely aligned by MUMmer v3.0, and mismatches at nucleotide level were found as mostly heterozygotes of the sequenced line itself. Finally, we chose PacBio/Hi-C versions of PF40 and PC02, and Illumina version of PC99, for all downstream comparative analysis. Sequencing gaps of PF40 (n = 180) and PC02 (n = 342) were uniquely aligned and filled by the corresponding Illumina sequences using BLASTN.

**Flow cytometry and K-mer analysis.** Flow cytometry[65] (CyFlow Cube, Partec, Germany) was used to estimate genome size of the allotetraploid PF40. Fresh young leaves (30–50 mg) of PF40 were finely chopped with a razor blade in buffer of CyStain Absolute T. After extraction, the solution was filtering through 30 μm nylon meshes, then 50 μl of RNase and propidium iodide (PI) were added immediately. Rice (*Oryza sativa* sp. Japonica Nipponbare, 394.6 Mb) was prepared following the same procedure as reference, and mixed with perilla extracts. Signals were detected with an air-cooled argon laser (Uniphase) at 488 nm, 20 mW. Perilla genome size was estimated according to the equation: 1C nuclear DNA content = (1C reference genome size × peak means of perilla)/(Peak mean of reference). We estimated genome sizes of the three perilla lines using K-mer frequency analysis with a K-mer size of 91 following published protocol[66].

**Evaluation of assembly quality.** We evaluated assembly completeness using BUSCO[67] v3.02 under genome mode (Supplementary Table 8). Expressed sequence tags (ESTs) downloaded from GenBank (as of 1 Oct, 2019) and published perilla RNA-seq transcripts[12,17] were mapped onto the PF40 genome using BLASTN with default parameters. Raw Illumina paired-end reads were mapped onto each cognate genome assembly using BWA[68] v0.7.10-r789.

**Repeat and gene annotation.** Repetitive sequences of the three perilla genomes were identified by a combination of homology-based and de novo approaches. Tandem repeats were predicted using Tandem Repeats Finder[69] v4.07b. For transposable elements, we first used RepeatMasker with the Repbase[70] v21.04 database of known repeats to search for transposable elements in the genomes, then RepeatProteinMask v4-0-6 was used by aligning the genomes to known repeat protein database. RepeatModeler v1-0-8 was run with default parameters for de novo prediction. Finally, repetitive sequences identified by different methods were combined into the final repeat dataset (Supplementary Fig. 6 and Supplementary Data 1). LTR-RTs were further identified by LTR_retriever[71]. Since direct repeats of a newly inserted LTR-RT are identical to each other, we used this identity value to extrapolate the age of LTRs, and plotted them according to LTR correspondence between PFA and PC02 (Supplementary Fig. 7).

The ab initio gene predictions were performed with three programs, including Augustus v3.0.3, GenSan v1.0, and Glimmer v3.02. We further used annotated proteins from seven published plant genomes, including *Mimulus guttatus*, *Sesamum indicum*, *Solanum lycopersicum*, *Solanum tuberosum*, *Vitis vinifera*, *Brassica rapa*, and *Arabidopsis thaliana*, for homology-based gene prediction with GeneWise v2.2.0. Finally, we used two sets of RNA-seq assembly data downloaded from ref. [12] (de novo transcriptome assembly from four mRNA samples of perilla seeds at different developmental stages, with 54,079 transcripts) and ref. [17] (from whole transcriptome of red and green forms of perilla leaves with 54,500 and 54,445 transcripts, respectively), together with 5538 perilla ESTs downloaded from GenBank, for RNA-Seq-based gene prediction with Augustus v3.0.3. Combination of these results using EVidenceModeler[72] v1.1.1 generated high-quality annotations of the three genomes, which were then used for further comparative curation. The final gene annotation dataset was evaluated using BUSCO under gene mode (Supplementary Table 8). To diagnose reasons of low BUSCO performance, where numbers of 'missing BUSCOs' were relatively high, we retrieved missing models from embryophyta_odb9 database and searched against the three perilla genomes. Hits of predicted pseudogenes were then checked for presence of coding SNPs or Indels which introduced premature stop codons or frameshifts, and the corresponding genotypes of these variants were retrieved. Gene functions were annotated according to the BLASTP best match (1e−8) against the Swiss-Prot v20160809 and TrEMBL Release 2016_08 databases. For annotation of non-coding RNAs, tRNA genes were predicted using tRNAscan-SE[73] v1.3.1, rRNA fragments were identified by aligning the assembled genomes with plant ribosomal RNA sequences using BLASTN, and microRNA/snRNA genes were detected using INFERNAL[74] v1.1rc2 against the Rfam[75] v12.1 database (Supplementary Tables 10 and 11).

**Subgenome partition and comparative analysis.** Illumina sequencing data of PC02 was mapped onto PF40 genome with BWA after Q30 filtering, and coverage depth in 5-kb window was calculated. With the missing of BB diploid information,

segments of AA subgenome origin were determined when its coverage depth was higher than half of genome peak depth (0.5 × 28 = 14). At least five contiguous segments of the same origin were required to define a confident PFA/PFB block, otherwise it will be classified as the opposite origin. Then PFA/PFB junction intervals were extracted to align with the PC02 genome using BLASTN for base-pair resolution. After subgenome partition, we constructed the orthology map between PFA, PFB, PC02, and PC99 segments by Mercator using annotated genes as input. The orthology map, consisting of 4030 orthologous blocks, was then used to guide nucleotide-level multiple alignments by MAFFT v7.271.

For comparative gene model curation, we first demarcated coding exons of the EVM-merged gene models along each of the four sequences, then scanned each MAFFT block from beginning to the end, to define the maximal interval spanning each orthologous prediction. Score of the top one BLASTP hit of each annotated gene against the NR database was retrieved. In most cases, only one predicted gene was observed along each sequence, and four orthologous gene models were evaluated in parallel. The prediction with highest BLASTP score was kept as guide gene model, and predictions from the remaining three sequences were discarded. When homologous proteins in NR were not available, BLASTN scores against the assembled RNA-seq transcripts were used. We mapped the guide gene model to the three gene-depleted intervals by GMAP version 2016-04-04 for re-annotation. When stop codons or coding frameshifts were encountered during exon projection, heterozygous SNPs, indels, or assembly errors of that focal genome were referred to for confirmation, resulting in pseudogene annotation of that interval. In rare occasions, two overlapping predictions were incorporated into one maximal interval, and the top two genes with highest BLASTP scores were evaluated for further GMAP analysis: if the two models overlapped on MAFFT alignment, only the one with higher score was kept for GMAP projection, otherwise both models were used.

For nucleotide mutation identification between PFA and PC02 after polyploidization, we cut the 4030 MAFFT blocks into separate intervals when contiguous alignment gaps of ≥100 bp in either sequence were observed, resulting in 128,106 intervals with a minimal length of 500 bp. Intervals with overall identity across the four sequences <80% were further filtered out. Finally, 42,022 conserved intervals with a total length of 123,601,930 bp were scanned for mutation identification by custom Perl scripts. Any nucleotide position with 1:3 genotype count was documented as one mutation and three ancestral nucleotides. The focal mutation must be flanked by five identical nucleotides on either side, to rule out false positive identification by erroneous alignment (Supplementary Fig. 13).

**Gene family analysis.** A total of 325,464 protein-coding genes from seven plant species with sequenced genomes and the four perilla genomes were used for gene family analysis (Supplementary Data 2). We used BLASTP to generate pairwise protein alignments with E-value threshold of 1e−8, then OrthoMCL[76] v2.0.8 was used to cluster similar genes with default parameters (inflation value I = 2.0). This resulted in 33,403 gene families comprising 280,203 genes from these 11 genomes (Supplementary Table 12). A phylogenetic tree was then constructed with 1057 1:1:1 single-copy orthologous genes using MrBayes[77] v3.2.1 with the general time-reversible model. Divergence times of these species were estimated using MEGA-CC[78] v7.00-2 with potato-tomato split time for calibration[79] (7.67 Mya).

**Genome synteny analysis.** To analyze chromosomal evolution of perilla during recent and ancient polyploidy history, we first identified orthologous genes using all-against-all BLASTP (1e−8). Homolog signals within 50 kb were taken as tandem duplications and filtered out, and syntenic blocks were detected with at least five collinear anchor genes using DAGchainer[80] r02-06-2008. Comparative gene curation information was further used to polish the synteny map. A total of 15,170 orthologous gene pairs were identified between PFA and PFB (Fig. 1b) and shown by Circos[81]. Collinear genes for PFA/PC02, PFB/PC02, and PC02/PC02 were 19,412, 15,422, and 1812 pairs, respectively, and displayed as dot-plots (Fig. 2b and Supplementary Fig. 12). These syntenic gene pairs were aligned using MUSCLE[82] v.3.8.31, and dN/dS of each gene pair was calculated using Codeml of the PAML[83] package v4.8. For dN/dS calculation involving PC99, orthologous relationship from comparative gene curation was used. We used the formula (1)

$$t = dS/2r \qquad (1)$$

for independent divergence time estimation, where r is the neutral substitution rate.

**Germplasm collection and resequencing.** A total of 191 tetraploid perilla accessions were collected in 2010–2018 (Supplementary Data 7), and maintained in Guizhou Academy of Agricultural Sciences. Each accession was planted with seeds from a single parental line in six square meters in Kaiyang county, Guizhou Province (26°56′N, 106°55′E). Genomic DNA were extracted following standard protocol for sequencing library construction with insert sizes of ~350 bp. An average 16× coverage of the assembled PF genome (~20 Gb) was generated on the Illumina HiSeq 2000 platform with PE150 mode for each accession.

Paired-end sequencing reads of each line were aligned to the PF genome simultaneously using BWA v0.7.10-r789 with default parameters, and only uniquely mapped reads were kept. PCR duplicates were marked using Picard v1.119 and indexed using the SAMtools package[84] v1.1. We used GATK[85]

v3.5-0-g36282e4 to infer SNPs and Indels. SNPs were filtered out with the following parameters:
'QD < 2.0 || MQ < 40.0 || FS > 60.0 || HaplotypeScore > 13.0 || MQRankSum < −12.5 || ReadPosRankSum < −8.0', and Indels were filtered out with the following parameters:
'QD < 2.0 || ReadPosRankSum < −20.0 || InbreedingCoeff < −0.8 || FS > 200.0 || SOR > 10.0'. Passed variants were annotated with ANNOVAR[86] ver.20111120 and SnpEff[87] v2.0.5.

Analysis of tetraploid resequencing data against the diploid genome will be compromised by cumulative mapping of both PFA and PFB reads. For this reason, we retrieved Illumina reads mapped onto PFA sequences of PF40 from BAM file, and re-mapped them onto PC02 genome. Five representative perilla lines from each of the three clades were used for variant calling by GATK. Totally 1,690,046 fixed mutations (shared by all of the 15 samples) were identified, and 6,599,943 SNPs (varied across these lines, not completely fixed in tetraploid yet) were used to extrapolate their divergence with the diploid (Supplementary Fig. 15d).

**Population analysis**. A subset of 4,789,738 high-quality SNPs (SNP quality ≥ 2000, minor allele frequency ≥ 0.05, and missing data ≤10%) from the entire SNP dataset was used to build a neighbor-joining tree in PHYLIP[88] v3.5c with 100 bootstrap replicates (Fig. 4a). We conducted PCA to evaluate genetic structure using EIGENSOFT[89] v6.0.1, and the first two eigenvectors were plotted in 2D (Fig. 4b). STRUCTURE[90] v2.3.4 was then used to infer the population structure with $K$ values from 2 to 4 by using 10,000 iterations (Fig. 4c). Linkage disequilibrium was analyzed using Haploview[91] v4.2 for the three clades, and LD decay was calculated on the basis of the $r^2$ value and the corresponding distance between two SNPs (Supplementary Fig. 16). Nucleotide diversity (π) of the three perilla clades was calculated individually to estimate degree of variability within each group, and fixation statistic $F_{ST}$ across the three clades was applied to explain population differentiation on the basis of the variance of allele frequencies between two groups. These two statistics were calculated with VCFtools[92] v0.1.15 using window size of 20 kb by 10-kb sliding.

**Analysis of homeologous exchange**. Homeologous exchanges (HEs) were replacements of chromosomal regions with duplicated copies from the homeologous sequences, which frequently resulted in duplication/deletion scenario of sequencing depth distribution between homeologs. Availability of syntenic map of the tetraploid perilla and the homeologous relationship of PFA/PFB genes facilitated HE identifications at segmental and genic levels. We first inferred coverage depth data of each resequencing sample in 5-kb window with uniquely mapped paired-end reads, and intervals with average depth higher than 1.5× and less than 3× were kept as candidate duplications (× is the genome average depth). Depth of the corresponding homeologous intervals based on syntenic map, when <0.5×, will be defined as deletion, and therefore a candidate HE event, from that accession. A minimal size of 20 kb (duplication or deletion interval) was required to define each HE. For HEs small than 20 kb, we focused on homeologous genes of PFA/PFB following the same procedure. Balanced HEs were identified based on coverage depth data of PC02 onto PF40 when PFA segments were flanked by long tract of PFB sequences on either side and vice versa.

**Transcriptome analysis**. Leaves and flowers of the three reference lines PF40, PC02, and PC99 were harvested to extract total RNA, and cDNA libraries were constructed using TruSeq RNA Library Prep Kit (Illumina). A total of 5 Gb clean data were generated for each sample on the Illumina HiSeq 2000 platform using the Illumina RNA-seq protocol. Three biological replicates were used for each sample. RNA-seq reads were mapped to the genomes using Tophat[93] v2.0.8, and the gene expression level was calculated using TPM (Transcripts Per Million) with RSEM[94] v1.3.2. Homeologous expression bias was analyzed by comparing the TPM values of syntenic PFA/PFB gene pairs, including both intact functional genes and annotated pseudogenes (n = 15,484). For any paired TPM values of <1, a pseudocount of 1 was added for both PFA and PFB values before log2 ratio calculation (Fig. 2d). Seven transcriptomes corresponding to the seeds of 2, 6, 10, 14, 18, 22, and 26 days post anthesis (DPA) of the high oil content accession PF40 (~56%) were sequenced and analyzed following the same protocol for elucidation of the transcriptional activities of fatty acid biosynthesis and TAG assembly related genes (Fig. 6a).

**Identification of genes involved in TAG biosynthesis and meiotic crossover**. We manually annotated genes related to fatty acid biosynthesis and triacylglycerol assembly in perilla genomes by comparing with the Arabidopsis acyl-lipid metabolism database[95] (Supplementary Data 8). To identify genes involved in meiotic crossover pathway, we retrieved eight Arabidopsis homologous proteins[53] from TAIR10.31, and searched against the perilla assemblies (Supplementary Data 9).

**Leaf trichome isolation**. Glandular trichomes are specialized epidermal cells where volatile oils and other secretions are synthesized and stored. We extracted total RNAs from leaf trichomes to identify candidate terpene synthase genes (TPSs) and cytochrome P450 genes (CYPs) involved in essential oil biosynthesis (Supplementary Data 10). Briefly, 10–15 g of young leaves ~1 cm wide was collected in 50 mL

centrifuge tube and covered with 40 mL of ice-cold wash buffer (50 mM Tris-HCl, pH 7.5, 200 mM sorbitol, 20 mM Suc, 10 mM KCl, 5 mM MgCl₂, 0.5 mM K₂HPO₄, 5 mM succinic acid, 1 mM EGTA, diethyl pyrocarbonate-treated water, 1 mM aurintricarboxylic acid, and 14 mM β-mercaptoethanol) for 15 min. Ten grams of 0.5-mm glass beads (Solarbio) was added to 50 mL centrifuge tube. The centrifuge tube containing the glass beads and leaves was shaken by hands 300 times followed by 60 s on ice, repeated two more times. The leaf slurry was then poured through a series of plastic funnels, each with an attached nylon mesh cloth with different pore sizes. The flow-through was collected in an ice-cold 50 mL centrifuge tube at each step. The order of these meshes was as follows: 100/200/300/400 mesh, with the peltate trichomes collected on the 300 mesh and the head trichomes collected on the 400 mesh. The trichome fractions were immediately processed for total RNA extraction, RNA-seq library construction, and Illumina sequencing.

**GWAS analysis**. High-quality SNPs from population resequencing (n = 4,789,738) were used to perform GWAS for leaf color variation and seed oil ALA content using FaST-LMM[96]. The genome-wide significance thresholds were evaluated with the formula $P = 0.05/n$, which were ~$1.04 \times 10^{-8}$. Perilla has three kinds of leaf colors: red, green, and abaxial red with adaxial green. There is no perilla leaf of adaxial red with abaxial green, implying existence of an independent factor for anthocyanin polar transport[97]. For this reason, leaf color phenotype was determined by visual inspection of the abaxial side only. Seed oil ALA contents were quantified by gas chromatography[12]. Briefly, perilla seeds were crushed and transmethylated at 105 °C for 120 min, then 0.5 g powder was mixed with 5 mL petroleum ether:ether mix solution (v/v 1:1). After transmethylation, 1.5 mL of 0.9% NaCl solution and 1.5 mL of n-hexane were added for production of fatty acid methyl esters, which were then analyzed on GC-2010 Plus Gas Chromatograph (Shimadzu, Japan) with one 30 m × 0.25 μm (inner diameter) HP-FFAP column (Agilent, USA), during which the oven temperature was increased from 170 to 180 °C at 1 °C/min.

**Reporting summary**. Further information on research design is available in the Nature Research Reporting Summary linked to this article.

## Data availability

The data supporting the findings of this work are available within the paper and its Supplementary Information files. A reporting summary for this article is available as a Supplementary Information file. The raw sequence reads, genome assembly, and gene annotation of PF40, PC02, and PC99 have been deposited in NCBI under the BioProject accession numbers PRJNA431002, PRJNA431004, and PRJNA431006, respectively. Source data are provided with this paper.

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

## Acknowledgements

This project was supported by National Natural Science Foundation of China (U1812403-1 to S.C.). Q.S. was supported in part by National Natural Science Foundation of China (31860391) and Guizhou Province (Qian Talent Project [2019]5656). We thank Prof. A. Liu and Dr. J. Xu for helpful discussions, X. Zhang and W. Wu for bioinformatics platform support, Z. Shang, S. Yang, J. Xu, H. Wen, and X. Wang for field assistance. The Hi-C sequencing and analysis were performed with help from Annoroad Gene Technology (Beijing).

## Author contributions

Y.Z. conceived and designed the project, D.Z. and Q.S. prepared samples and extracted DNA/RNA for genome and transcriptome sequencing, L.L. and Z.N. generated the genome assemblies, Y.Z. and L.L. annotated the genomes and analyzed population data, Q.S. maintained the perilla germplasm resources, D.Z., Sha C., and Y.S. performed wet-lab molecular experiments, Y.Z. and S.C. jointly supervised the project, Y.Z. wrote the paper. All authors read and approved the manuscript.

## Competing interests

The authors declare no competing interests.
