## [Peer Review File · Nature Communications]

Incipient diploidization of the medicinal plant *Perilla* within 10,000 yearsREVIEWER COMMENTS

Reviewer #1 (Remarks to the Author):

Perilla can be used as food and medicine for its rich nutritional values. The incipient diploidization of Perilla is also an interesting topic in the plant evolution. In this manuscript, the authors reported a high-quality reference genome sequence, found some interesting events in evolution and identified two candidate genes underlying agronomic traits (color and oil) through GWAS approach, which provide a foundation for future genetic researches on Perilla.

The genome analyses in the paper are nearly complete and rich enough, and I only have three suggestions for the work:

1. The validation experiments on the candidate genes are very weak in the paper. At least, expression profiling of the genes should be performed for a few accessions with different alleles. Solid biological results are needed here.

2. In the population-structure analysis, the authors divided all the accessions into three clades, and I'm not clear whether all "crispa" are from Southeast Asia (see Fukushima et al., 2015). For "We selected 191 tetraploid accessions ... from China and abroad ...", what about the geographic origin of the accessions? The descriptions on the accessions should be made clearer, with more details provided.

Fukushima, A., Nakamura, M., Suzuki, H., Saito, K. & Yamazaki, M. High-throughput sequencing and de novo assembly of red and green forms of the *Perilla frutescens* var. *crispa* transcriptome. *PLoS One* 10, e0129154 (2015)

3. The transcriptome data and the phenotypic data of the accessions should be released as well.

Reviewer #2 (Remarks to the Author):

This manuscript describes the assembly, annotation, and analysis of the perilla genome. An allotetraploid as well as a diploid progenitor was sequenced along with a diversity panel. The authors focused on characterizing the polyploidization event and its impact on the genome, the fractionation of the tetraploid genome, population structure in perilla, and identifying loci associated with pigmentation and oil content. Overall, the authors performed a wide set of analyses and generated a robust dataset for mining this genus.

Specific comments:

The manuscript would be greatly improved by editing to improve flow, clarity, and conciseness.

The supplemental tables and figures should be cited in order; this was very distracting to find the tables and figures out of order

Titles and appropriate legends are needed for the supplementary tables

The two subgenomes within the PF genome were identified by PC02 read mapping. This is reasonable. It is surprising that some of the PF chromosomes 'appear' chimeric which to me is a bit surprising being the authors state this perilla underwent polyploidization 10,000 years ago. An equally likely scenario is that these PF chromosomes are chimeric with respect to the assembly. Additional evidence that the PF is correctly assembled is needed. It would be helpful if the discussion of the subgenomes via PC02 read mapping is done after intragenome synteny studies and the results of the two datasets

discussed.

The annotation methods are not optimal as indicated by the poor BUSCO scores of the genome annotation. Likewise, how 'pseudogenes' were identified is problematic. It has been well demonstrated in Arabidopsis that simple projection of gene models across accessions introduces substantial artifacts simply due to alternative splice forms.

Lines 131-132-the enrichment of biosynthetic genes involved in specialized metabolism in the PFB pseudogenes is not evidence for chemical diversity in the species. The authors did not look at variation within the species, they looked at 4 homeolog/orthologs.

Line 163: Since you do not know the B subgenome progenitor, it is incorrect to state that the BB-derived chromosomes underwent more rearrangements than the PFA subgenome.

It is totally unclear what is referred to when Illumina draft assemblies are mentioned. Are these individual draft assemblies of the resequenced perilla lines? If so, how good are these? Which genomes are being used in the analyses? If it is Illumina assemblies of PF40 and PC02-why do this?

Fig 2d. The figure itself is impossible to read but based on the associated text, it is not clear that there is an expression level difference between genes in subgenome A vs P. What statistics were performed to make this statement?

Line 217: How does the number of HEs detected compare to that seen in other nascent polyploids?

Line 509-517: The methods for the assembly of the Illumina genomes is woefully inadequate. What was assembled? The PF40 and PC02? The diversity panel? How were the Illumina assemblies benchmarked for quality? If it is of PF and PC02, why use an Illumina assembly when you have a PacBio assembly?

Line 570: The genome annotation methods are limited. What transcriptome data was assembled? How was it assembled? Why did you not use protein evidence from other Lamiaceae species?

Supp Fig 5: this can be deleted.

Supp Fig 11: Why not use some Lamiaceae genomes in this analysis to better show the position of perilla in the Lamiaceae?

Supp Fig 18: The microsynteny between Arabidopsis and Perilla is unclear from this figure. Remake the figure so that it is more apparent

Reviewer #3 (Remarks to the Author):

A nice study on the structure of a tetraploid genome, its origin and evolution. My comments are summarized in the attached file.

I am concerned about the dating of the WGD event. The authors came from 200 000 to 10 000 year, saying that the former estimate was an overestimate.OK, but the difference is 20-times.

I did not find Discussion comprehensive, comparing these newly acquired results with data available for other polyploid angiosperm genomes.

Reviewer #1

1. The validation experiments on the candidate genes are very weak in the paper. At least, expression profiling of the genes should be performed for a few accessions with different alleles. Solid biological results are needed here.

Author response:

Thanks for the comment. The red/green leaf color variation and the high ALA content of seed oil are two major agronomic traits of perilla, and candidate genes were identified by GWAS using high-density polymorphism data of the resequencing population. While these results were solid from perspective of genome research, we agreed that further experimental validations are needed to elucidate details of these regulatory or metabolic pathways. Following your kind suggestions, we extracted RNAseq data of the candidate genes from related samples of different genotypes, and reported in Supplementary Fig.18. In-depth analysis of the two traits by genetic and in vitro biochemical assays are underway in our lab.

2. In the population-structure analysis, the authors divided all the accessions into three clades, and I'm not clear whether all "crispa" are from Southeast Asia (see Fukushima et al., 2015). For "We selected 191 tetraploid accessions ... from China and abroad ...", what about the geographic origin of the accessions? The descriptions on the accessions should be made clearer, with more details provided.

Author response:

Thanks for the comment. The *crispa* clade of perilla is characterized by wrinkled and curled leaves with rounded serrated edges, and had been reported occasionally from Japan, Korea, China, Myanmar, and Vietnam, suggesting a wide distribution from East to Southeast Asia. We had added this information in Line 256. In addition, geographic origin data of our germplasm collections was provided in details in Supplementary Table 23.

3. The transcriptome data and the phenotypic data of the accessions should be released as well.

Author response:

Thanks for the comment. The transcriptome data used in this manuscript had been released under PRJNA634235, and phenotypic data of these resequencing lines, including leaf color and seed oil ALA content, had been provided in Supplementary Table 23.

Reviewer #2

The manuscript would be greatly improved by editing to improve flow, clarity, and conciseness.

The supplemental tables and figures should be cited in order; this was very distracting to find the tables and figures out of order

Titles and appropriate legends are needed for the supplementary tables

Author response:

Thanks for the comment. We had revised numbering of tables and figures in their cited order, and added titles and necessary notes for supplementary tables. We had also revised the manuscript and methods wherever needed for more clarity and conciseness.

The two subgenomes within the PF genome were identified by PC02 read mapping. This is reasonable. It is surprising that some of the PF chromosomes ‘appear’ chimeric which to me is a bit surprising being the authors state this perilla underwent polyploidization 10,000 years ago. An equally likely scenario is that these PF chromosomes are chimeric with respect to the assembly. Additional evidence that the PF is correctly assembled is needed. It would be helpful if the discussion of the subgenomes via PC02 read mapping is done after intragenome synteny studies and the results of the two datasets discussed.

Author response:

Thanks for the comment. We agreed that chimeric recombination between PFA and PFB was a bit surprising. To rule out possibility of assembly artefacts, we compared the two versions of our assembly around chimeric junctions. Since the Illumina mate-pair assembly pipeline was independent from PacBio/HiC pipeline, consistency of the two versions confirmed authenticity of the rearrangement. In addition, genome-wide all-against-all Hi-C interaction map of PF40 also suggested correctness of the allotetraploid genome assembly. Finally, we designed PCR primers to validate all of the five PFA-PFB fusions (Supplementary Table 9), and all of these inter-subgenome rearrangements were confirmed as real.

Since the shortest fragment of inter-subgenome rearrangement was 586,506 bp with nine anchor genes (towards the long arm telomere of chr20, Supplementary Table 9), while synteny detection required at least five anchor genes, all of the PFA-PFB rearrangements had been covered by our synteny map. In addition, no HEs of the PF40 genome itself had been detected, suggesting that final results of subgenome synteny analysis by intra-genome synteny and diploid read mapping, regardless of their sequences, will be the same.

The annotation methods are not optimal as indicated by the poor BUSCO scores of the genome annotation. Likewise, how ‘pseudogenes’ were identified is problematic. It has been well demonstrated in Arabidopsis that simple projection of gene models across accessions introduces substantial artifacts simply due to alternative splice forms.

Author response:

Thanks for the comment. It’s well known that gene prediction programs are sensitive to coding-disruptive variations, including premature stop codons and indels. Since we had four syntenic (sub)genomes available, comparative annotation will improve gene model accuracy substantially, as had been done with four related *Saccharomyces* species (Nature 423:241-254). Otherwise, incongruence at start codon position, stop codon position, or exon boundary will result in extension, truncation, or split of gene models (Figures 4 and 5 in the *Saccharomyces* paper). After base-error corrections, the base-to-base orthologous relationship across the four subgenomes enabled us to spot these coding-disruptive variations unambiguously. By evaluating four syntenic gene models across each specific locus, we selected the best gene model for projection, and any projected models with coding-disruptive variants were defined as pseudogenes, which will otherwise be classified as fragmented partial genes. This procedure enabled us to measure the extent of subgenome fractionation properly.

As for drawbacks of our gene projection pipeline for alternative splicing, we didn’t focus on this type of annotation. In fact, we predicted only one model for each specific locus with the longest amino-acid sequence. For example, the TAIR10.1 release of *Arabidopsis* genome has 48,265

annotated genes from 25,113 unique loci, and only 23,559 genes were used for gene family analysis (Supplementary Table 14). We will analyze alternative splicing of perilla genes with regard to polyploidization in subsequent paper soon.

We also noticed the low performance of BUSCO evaluation of gene annotation, where numbers of “missing BUSCOs” were relatively high. We retrieved those missing BUSCO models from embryophyta_odb9 database and searched against the three perilla genomes. It turned out that 74, 105, and 82 of these missed BUSCO models had been annotated as pseudogenes in PF40, PC02, and PC99, respectively. In fact, annotated pseudogenes from GMAP projection, which contained stop codons within CDS, were not included in the final annotation files. This explained the BUSCO score problem.

Lines 131-132-the enrichment of biosynthetic genes involved in specialized metabolism in the PFB pseudogenes is not evidence for chemical diversity in the species. The authors did not look at variation within the species, they looked at 4 homeolog/orthologs.

Author response:

Thanks for pointing out this faulty assertion. We agreed that gene loss and pseudonization between subgenomes, which was inherited by all polyploid lines after WGD, contributed nothing to chemical diversity of the tetraploid population, and only variations within tetraploids were related to chemical diversity. We had deleted this expression.

Line 163: Since you do not know the B subgenome progenitor, it is incorrect to state that the BB-derived chromosomes underwent more rearrangements than the PFA subgenome.

Author response:

Thanks for the comment. We agreed that structural variations between PFA and PFB might have been fixed within parental diploids before polyploidization, and direction of these rearrangements was not self-evident without outgroup information. Following your suggestion, we had revised this sentence as “Large-scale variations of BB-derived chromosomes, especially chr2, chr6, chr16, and chr19, were observed when compared with PC/PFA. With the shortage of the BB diploid and outgroup information, directions and dating of these structural variations cannot be determined” in Line 160.

It is totally unclear what is referred to when Illumina draft assemblies are mentioned. Are these individual draft assemblies of the resequenced perilla lines? If so, how good are these? Which genomes are being used in the analyses? If it is Illumina assemblies of PF40 and PC02-why do this?

Author response:

Thanks for the comment, and we are sorry for this confusion. We initially chose the Illumina pipeline for assembly of PF40, PC02, and PC99 genomes by using paired-end, mate-pair, and 10X data. Unfortunately, it performed not well in the repeat-rich perilla genomes, with final contig N50s of less than 100 Kb. Then we re-assembled the PF40 and PC02 genomes by PacBio/HiC pipeline using the same perilla accessions, which finally improved sequence contiguity significantly (Supplementary Table 1). After evaluation, we chose PacBio versions of PF40 and PC02, and Illumina version of PC99, for further downstream analysis. We had clarified the procedures in method section.

Fig 2d. The figure itself is impossible to read but based on the associated text, it is not clear that there is an expression level difference between genes in subgenome A vs P. What statistics were performed to make this statement?

Author response:

Thanks for the comment. Subgenome expression bias of perilla was not clear at first glimpse, so we directly compared the number of PFA genes up-regulated (7,013) to that of PFB up-regulation (6,205), and coming to a conclusion of 13.0% excess of AA dominance. After careful inspection of results of PF40 RNAseq data from flowers and leaves (each with three replicates), a minor peak around 0.2 was noticed for all samples, suggesting slight up-regulation of PFA genes. We had revised this paragraph accordingly.

Line 217: How does the number of HEs detected compare to that seen in other nascent polyploids?

Author response:

Homeologous exchanges were identified by population resequencing and mapping, and the number of HEs should be dependent on both the number of lines sequenced and intrinsic HE frequency of the species. In *Brassica napus*, totally 17 HEs (14 from Cn to An and 3 from An to Cn) were identified from seven accessions (ref. 6). In nascent allotetraploid wheat, 37 HEs were reported from five individuals of three lineages (ref. 34). We identified totally 29 segmental HEs longer than 20Kb from 191 perilla lines.

Line 509-517: The methods for the assembly of the Illumina genomes is woefully inadequate. What was assembled? The PF40 and PC02? The diversity panel? How were the Illumina assemblies benchmarked for quality? If it is of PF and PC02, why use an Illumina assembly when you have a PacBio assembly?

Author response:

Thanks for the comment. We initially chose the Illumina procedure for assembly of PF40, PC02, and PC99 genomes by using paired-end, mate-pair, and 10X data. While the same assembly protocol by one of our co-authors had been successful in other genome projects (Cell (2012) 148:780–791, Nat. Genet. (2013) 45:456–461, and Nat. Genet. (2015) 47:625–631), it performed not well in the repeat-rich perilla genome, with final contig N50s of less than 100 Kb (Supplementary Table 5). Since the allotetraploid perilla was formed quite recently, detailed comparison of tetraploid and diploid is necessary to elucidate incipient diploidization at nucleotide, segmental, and chromosomal levels, yet low contiguity of these assemblies limited our analytical resolution. For this reason, we re-assembled the PF40 and PC02 genomes by PacBio/HiC procedures using the same perilla lines, which finally improved sequence contiguity significantly (Supplementary Table 1). Since these two pipelines were independent, we used the Illumina data for gap closing of the PacBio assemblies, and totally 180 (PF40) and 342 (PC02) gaps were uniquely filled. Further pairwise alignments of the two assembly versions by MUMmer v3.0 revealed that nearly all mismatches at nucleotide level were from heterozygosity of the sequenced line. Finally, we chose PacBio versions of PF40 and PC02, and Illumina version of PC99, for further analysis. We had revised this section accordingly for clarity.

Line 570: The genome annotation methods are limited. What transcriptome data was assembled? How was it assembled? Why did you not use protein evidence from other Lamiaceae species?

Author response:

Thanks for the comment. For homologue protein-based annotation, we used seven related plant genomes including *Mimulus guttatus*, *Sesamum indicum*, *Solanum lycopersicum*, *Solanum tuberosum*, *Vitis vinifera*, *Brassica rapa*, and *Arabidopsis thaliana*, where both *M. guttatus* and *S. indicum* were from Lamiaceae. Indeed, when we annotated the perilla genomes, only *M. guttatus* and *S. indicum* had gene annotation data available. That's the reason why no other Lamiaceae protein evidence was used for annotation. For RNAseq-based annotation, we didn't assemble transcriptome by ourself, but downloaded assembled RNAseq data from ref.12 (de novo transcriptome assembly from four mRNA samples of perilla seeds at different developmental stages, with 54,079 transcripts) and ref.17 (from whole transcriptome of red and green forms of perilla leaves with 54,500 and 54,445 transcripts, respectively). These transcripts, together with 5,538 downloaded perilla ESTs, were combined together for gene annotation. Accordingly, we revised annotation method as follows: "The ab initio gene predictions were performed with three programs, including Augustus v3.0.3, GenScan v1.0, and Glimmer v3.02. We further used annotated proteins from seven published plant genomes, including *Mimulus guttatus*, *Sesamum indicum*, *Solanum lycopersicum*, *Solanum tuberosum*, *Vitis vinifera*, *Brassica rapa*, and *Arabidopsis thaliana*, for homology-based gene prediction with GeneWise v2.2.0. Finally, we used two sets of RNAseq assembly data downloaded from ref.12 (de novo transcriptome assembly from four mRNA samples of perilla seeds at different developmental stages, with 54,079 transcripts) and ref.17 (from whole transcriptome of red and green forms of perilla leaves with 54,500 and 54,445 transcripts, respectively), together with 5,538 perilla ESTs downloaded from GenBank, for RNASeq-based gene prediction with Augustus v3.0.3" (Lines 602 to 611).

Supp Fig 5: this can be deleted.

Author response:

Thanks for the comment. We had deleted this GC-depth evaluation result.

Supp Fig 11: Why not use some Lamiaceae genomes in this analysis to better show the position of perilla in the Lamiaceae?

Author response:

Thanks for the comment. There are totally 12 genome assemblies from 8 Lamiaceae species available now, yet only four of them (*Salvia miltiorrhiza*, *Salvia bowleyana*, *Scutellaria barbata*, and *Scutellaria baicalensis*) had gene annotation data. For a better elucidation of perilla phylogeny, we updated the phylogenetic tree by incorporating four Lamiaceae genomes, including *Salvia miltiorrhiza* (Genus *Salvia*), *Scutellaria baicalensis* (Genus *Scutellaria*), *Mimulus guttatus* (Genus *Mimulus*), and *Sesamum indicum* (Genus *Sesamum*). It turned out that Genus perilla was close to *Salvia* (Supplementary Fig.9).

Supp Fig 18: The microsynteny between Arabidopsis and Perilla is unclear from this figure. Remake the figure so that it is more apparent

Author response:

Thanks for the comment. We had re-drawn this figure by keeping AT1G and AT5G only. Micro-synteny between the two Arabidopsis segments was clear with perilla sequence as bridge (Supplementary Fig.17).

Reviewer #3

A nice study on the structure of a tetraploid genome, its origin and evolution. My comments are summarized in the attached file.

I did not find Discussion comprehensive, comparing these newly acquired results with data available for other polyploid angiosperm genomes.

Author response:

Thanks for the comment. We agreed that the Discussion was a bit superficial. We have revised this section by reviewing results from published polyploid genomes of recent origin, and compared with our new findings from perilla (Lines 346 to 370).

Information on genome size of the PC genome is missing.

Author response:

The assembled PC02 and PC99 genomes was 676.9 and 618.8 Mb, respectively (Supplementary Tables 1 and 5). Previously we only analyzed PF40 by flow cytometry. Now we have analyzed all of the three samples (Supplementary Fig.2). Genome sizes from sequence assembly, K-mer size estimation (Supplementary Fig.1), and flow cytometry agreed quite well.

Pg. 4 „Notably, these disabled PFB genes were enriched for biosynthesis of secondary metabolites ($P = 1.72 \times 10^{-6}$, Fisher's exact test), implying genetic basis of chemical diversity of this herbal species11”

I was not sure how disabled genes can be enriched for...? I do not understand “genetic basis of chemical diversity” what are other possible bases of chemical diversity?

Author response:

Thanks for pointing out this faulty assertion. We agreed that gene loss and pseudonization between subgenomes, which was inherited by all polyploid lines after WGD, contributed nothing to chemical diversity of the tetraploid population. We had deleted this expression during revision (Line 128).

Pg. 4, bottom. “Phylogenetic tree with 1,057 single copy orthologous genes suggested that the Perilla genus was closely related to Mimulus,...”

It was confusing to read about Mentha (Lamiaceae) and then to read that Perilla was closely related to Mimulus....Should not be mint or some other Lamiaceae species/genera closer to Perilla than Mimulus?

Author response:

Thanks for the comment. We cited the heterozygosity data of Mentha (Line 87), an out-crossing mint species, to highlight the selfing nature of perilla. Unfortunately, gene annotation data of Mentha is not publicly available, and cannot be used in phylogenetic tree construction. Indeed, there are totally 12 genome assemblies from 8 Lamiaceae species available now, yet only four of

them (*Salvia miltiorrhiza*, *Salvia bowleyana*, *Scutellaria barbata*, and *Scutellaria baicalensis*) had gene annotation data. For a more comprehensive phylogeny of perilla, we updated the phylogenetic tree by incorporating *Salvia miltiorrhiza* (Genus *Salvia*), *Scutellaria baicalensis* (Genus *Scutellaria*), *Mimulus guttatus* (Genus *Mimulus*), and *Sesamum indicum* (Genus *Sesamum*) data. It turned out that Genus perilla was closer to *Salvia* than to *Mimulus* (Supplementary Fig.9).

And further on: “a yet unknown BB donor” – does that mean that the genus Perilla has more than two species as stated in Introduction? Please clarify.

Author response:

Theoretically, the *Perilla* genus should have at least three species, including the AABB allotetraploid, the AA diploid donor, and the BB diploid donor. As we showed in Supplementary Fig.5, all of our diploid germplasm collections were from the same AA donor, while diploid species that can be mapped onto PFB segments is not available yet. In fact, it had long been known that “there is still incomplete information on the second diploid genome donor” of perilla (Natural Medicines (1994) 48:185-190, Genetic Resources and Crop Evolution (2005) 52:797–804). Endeavors to find the BB diploid perilla from its presumed wild habitat, mountainous terrains of the Yangtze River region, are now underway.

I am concerned about the dating of the WGD event. The authors came from 200 000 to 10 000 year, saying that the former estimate was an overestimate. OK, but the difference is 20-times.

Fig. 5 “Indeed, 577 out of the 1,057 single copy orthologous genes had no synonymous substitutions either, implying that molecular dating by concatenating coding sequences of single copy genes overestimated polyploidization time in this extreme scenario²³. Compared with the ~ 7500-year-old allopolyploid *Brassica napus* where 18.6 % genes were identical between tetraploid and diploid progenitor⁷, the allotetraploid *P.frutescens* should have formed post Neolithic within 10,000 years.”

Whereas I understand that the estimate based on single copy genes can be overestimated, I was not able to follow the subsequent conclusion about “within 10,000 years”. I did not analyze whether this conclusion is correct or justified, however I was not able to follow the argumentation presented herein.

Author response:

Thanks for the comment. Phylogenetic dating is usually done with single copy orthologous genes across related genomes. In nascent polyploids, both of the two parental genes should have been kept coding-intact, so we take PFA and PFB of the allotetraploid as independent species for convenience. A caveat here is that, under relaxed selective pressure on homeologs, divergence times of AA-BB and PFA-AA will be overestimated to some extent. We updated this analysis by incorporating the AACC *Brassica napus* and both of its diploid progenitor genomes (Supplementary Fig.9). Result suggested that divergence time of PFA-PC02 was 0.244 Mya, which was one third of that of *Brassica napus*-*Brassica oleracea* (0.775 Mya, based on CC subgenome). This implied that tetraploid perilla should be younger than *Brassica napus*. A better estimation of polyploidization time was direct comparison of PFA-PC02 syntenic orthologous genes. This method had been applied in the original *Brassica napus* paper for age estimation, where dS peak around $2.5e-4$ was observed (Figure S14 in ref. 6). A divergence time of

7,500-12,500 years was then calculated for Brassica napus. In perilla, however, totally 49.1% genes between PFA-PC02 had no synonymous substitutions at all, resulting in exponential decay of Ks distribution plot with no peak. Proportion of identical genes between tetraploid and diploid was higher in perilla (30.9%) than in Brassica napus (18.6%), also suggesting a more recent origin of perilla. Taken together, we propose that tetraploid perilla should be younger than Brassica napus, formed within the recent 10,000 years.

Line 163: „The BB-derived chromosomes, especially chr2, chr6, chr16, and chr19, had undergone much more rearrangements than PFA.” I am not if I understand how you knew about structure of the B genome. What makes you think that A and B genomes had very similar structure? If I misunderstood something here (perhaps I do not understand what you mean by “much more rearrangements than PFA”), please modify the current wording.

Author response:

Thanks for the comment. We agreed that structural variations between PFA and PFB might have been fixed within parental diploids before polyploidization, and direction of these rearrangements was not self-evident without outgroup information. Following your suggestion, we had revised this sentence as “Large-scale variations of BB-derived chromosomes, especially chr2, chr6, chr16, and chr19, were observed when compared with PC/PFA. With the shortage of the BB diploid and outgroup information, directions and dating of these structural variations cannot be determined” in Line 160.

Pg 5, line 168

detailed sequence alignments with Illumina draft assemblies suggested 17 out of 19 chromosomal inversions were authentic (Supplementary Table 16). Three bona fide inversions of PC segments were observed with the largest one of 14.8 Mb (Supplementary Fig. 12), suggesting that the diploid donor species was also in dynamic karyotype evolution since polyploidization.

Here a reader should better understand how you inferred the direction of inversions, i.e. how you decided whether an inversion was in the parental genome or PF genome or vice versa?

Author response:

As we just discussed, when rearrangements were observed on PFB, we cannot determine the directions and timing of these events unambiguously. However, if variations occurred on PFA, both PFB and PC can be taken as outgroup, and inversions can be confirmed definitely as occurring after WGD under the maximum parsimony scenario. Indeed, we had used the same rationale in nucleotide mutation direction analysis (Supplementary Fig.13). Situations were the same for inversions on PC chromosome, where both PFA and PFB were used as outgroup, and the longest PC inversion was validated by micro-synteny with Arabidopsis (Supplementary Fig.11b). Accordingly, we put a footnote in Supplementary Table 16 as follows:

1. Consistency of PacBio sequence with Illumina raw assembly confirmed authenticity of these inversions as real, rather than as assembly artefacts.
2. Inversions on PFA were determined by comparing with PC and PFB segments.
3. Inversions on PC were determined by comparing with PFA and PFB segments.
4. Directions and timing of those seemingly PFB-located inversions cannot be determined definitely.

“Further analysis of PC02 genome revealed 66 scattered duplication blocks involving 1,812 pairs of genes (Supplementary Fig. 13), representing relics of ancient whole-genome duplication history of the diploid. The calculated Ks distribution peak around 0.9 had been observed in most Lamiaceae species^{24,25}, suggesting occurrence of polyploidization at the basal lineage of Lamiaceae about 68 Mya²⁴.”

Could the Lamiaceae-specific WGD be confirmed also for the B genome based on the A-B synteny?

Author response:

Thanks for the comment. Since the three perilla genomes (PFA, PFB, and PC) were highly syntenic, Lamiaceae-specific WGD signals were also observed within PFA and PFB subgenomes, respectively. We had provided an example of dot-plot across six homologous segments in Supplementary Fig. 12b.

Pg 7: “indicated balanced swap of homeologous segments rather than random recombination in polyploid.”

I am not sure what the authors aim to say here. Would random recombination (whatever this is) generate a reciprocal translocation?

Author response:

Thanks for the comment. What we wanted to say was “balanced reciprocal translocation between homeologs”. We had corrected this paragraph as “we identified 18 balanced reciprocal translocations between syntenic PFA and PFB homeologs” in Line 230.

Pg. 7, line240:

Aneuploidy of c. 35 Mb segments. So, how many 35 Mb segments in total?

Figure 3c shows a reciprocal translocation. This whole section is unclear. What is aneuploidy in your definition? An entire chromosome missing? Does this mean that a 35 Mb segment (from A genome) was not identified in the PF genome?

Author response:

Thanks for the comment. We agreed that aneuploidy was inappropriate here since chromosome number was not affected. Major effect of these translocations was gene dosage imbalance, and we had corrected this paragraph as “Two of these events resulted in gene dosage imbalance of ~35 Mb segments” in line 237.

REVIEWER COMMENTS

Reviewer #1 (Remarks to the Author):

For the response to reviewer #1, I'm satisfied with the revision.

For the response to reviewer #2, I have one remaining concern --

Reviewer #2 noticed that "The annotation methods are not optimal as indicated by the poor BUSCO scores of the genome annotation".

The authors responded that "numbers of missing BUSCOs were relatively high" and "74, 105, and 82 of these missed BUSCO models had been annotated as pseudogenes in PF40, PC02, and PC99, respectively".

The authors should carefully check whether the stop codons within CDS of the missing BUSCOs resulted from heterozygous SNPs or assembling errors. Moreover, discussions on the poor BUSCO scores of the genome annotation should be added in the revision.

No other concerns for the paper.

Reviewer #3 (Remarks to the Author):

As the authors satisfactorily responded to my questions and criticism, I am pleased to recommend the revised manuscript for publication.

Reviewer #1

1. For the response to reviewer #2, I have one remaining concern --

Reviewer #2 noticed that "The annotation methods are not optimal as indicated by the poor BUSCO scores of the genome annotation". The authors responded that "numbers of missing BUSCOs were relatively high" and "74, 105, and 82 of these missed BUSCO models had been annotated as pseudogenes in PF40, PC02, and PC99, respectively".

The authors should carefully checked whether the stop codons within CDS of the missing BUSCOs resulted from heterozygous SNPs or assembling errors. Moreover, discussions on the poor BUSCO scores of the genome annotation should be added in the revision.

Author response:

Thanks for the comment. Following your kind suggestion, we manually checked presence of coding SNPs or Indels for these predicted pseudogenes which should introduce premature stop codons or frameshifts during comparative gene model curation, and the corresponding genotypes of these variants were also retrieved. It turned out that among the 74 annotated pseudogenes in PF40, 25 were caused by polymorphic (heterozygous) SNPs or Indels, and 49 were caused by fixed variations. Similar scenarios were also observed in PC02 and PC99 (Supplementary Table 8). Presence of coding Indel turned out to be a major reason for pseudogenization, accounting for about 84% of these instances. In addition, no assembling or sequencing errors were observed, confirming validity of our conclusion.

We had updated this result in the main text (Lines 128-133) and method (Lines 617-622). Breakdown of these retrieved pseudogenes by four types of variations was also attached in Supplementary Table 8.

REVIEWERS' COMMENTS

Reviewer #1 (Remarks to the Author):

I'm satisfied with the revision now.